

# Quantifying Small-scale Temperature Variability using Distributed Temperature Sensing and Thermal Infrared Imaging to Inform River Restoration

Jessica R. Dzara[1], Bethany T. Neilson[1], Sarah E. Null[2]

[1]Department of Civil & Environmental Engineering, Utah State University, 8200 Old Main Hill, Logan, Utah, 84321-8200, USA
[2]Department of Watershed Sciences, Utah State University, 8200 Old Main Hill, Logan, Utah, 84321-8200, USA

*Correspondence to*: Jessica R. Dzara (woodjessicarose@gmail.com) and Sarah E. Null (sarah.null@usu.edu)

Keywords: stream temperature, temperature range, thermal refugia, trout, simulation, Walker River

**Abstract.** Watershed-scale stream temperature models are often one-dimensional because they require less data and are more computationally efficient than two- or three-dimensional models.  However, one-dimensional models assume completely mixed reaches and ignore small-scale spatial temperature variability, which may create temperature barriers or refugia for cold water aquatic species.  Fine spatial- and temporal-resolution stream temperature monitoring provides information to identify river features with temperature ranges that differ from the reach average. We used a distributed temperature sensing system to

observe small-scale stream temperature variability, measured as temperature range through space and time, within two 400 meter reaches in summer 2015 in Nevada's East Walker and mainstem Walker Rivers.  In addition, thermal infrared aerial imagery collected in summer 2012 quantified the spatial variability of river temperatures throughout the Walker Basin.  Both the distributed temperature sensing data and thermal infrared aerial imagery were used to corroborate prior temperature model results.  Additionally, these data highlighted that beaver dams and irrigation return flow channels maximize thermal variability

and can provide thermal refugia, while groundwater seeps provide small cooler areas and diversion canals often create warm local temperatures downstream.  To extend temperature predictions and obtain a better understanding of thermal variability at the watershed-scale, temperatures bounds from observations by river features were added to the longitudinal temperature predictions. These results show that while bulk stream temperatures are often too warm to support trout and other cold-water species, thermal refugia may exist to improve habitat connectivity and passage for migratory species between Walker River

and Lake.  Overall, river restoration efforts should focus on maintaining and enhancing features that create this thermal variability and habitat connectivity.



## 1 Introduction

Trout and salmon avoid heat stress by sheltering in pockets of cold water when stream temperatures are near upper thermal tolerances (Dunham et al. 2003; Sutton et al. 2007). Assessing stream temperatures at small temporal and spatial scales is thus important to quantify stream temperature heterogeneity and best manage or restore complex and variable riverine habitats (Vatland et al. 2015). However, one-dimensional stream temperature models applied at the watershed-scale are poor predictors of thermal micro-habitats. High resolution temperature monitoring provides micro-habitat information, but is typically conducted over small spatial extents and thus difficult to extrapolate to the watershed scale for management and restoration decisions.

Stream temperature models are a useful tool for river management, helping decision makers to understand stream temperature dynamics and the potential impact of restoration and management. Many one-dimensional temperature models exist, and have been applied to understand temperature effects of dams, reservoir re-operation, climate change, and restoration in systems all over the world (Bond et al., 2015; Elmore et al., 2015; Pelletier et al., 2006). Stream temperature models used in management are often one-dimensional because they are less data intensive and more computationally efficient than two- or three-dimensional models; however, one-dimensional models assume perfectly mixed reaches. Thus, one-dimensional models do not identify small-scale features like cold water pools, cool margins or groundwater influenced areas, and reaches that are incompletely mixed laterally.

Stream temperature sensors estimate stream temperatures at fine spatial and temporal resolution at point locations. However, distributed temperature sensing (DTS) approaches provide near-continuous temperature measurements in both time and space (Selker et al. 2006; Suárez et al. 2011). Raman spectra DTS is capable of measuring temperatures every meter along a fiber optic cable with an accuracy of at least ±0.1 ºC (Tyler et al., 2009), and cables vary between approximately 1 – 10 km. In addition to quantifying thermal dynamics in air, streams, lakes, soil, and snow, DTS has determined zones of groundwater influence (Hare et al. 2015; Selker et al. 2006; Suárez et al. 2011) and hyporheic exchange (Briggs et al., 2012). Thermal infrared (TIR) remotely sensed imagery similarly captures spatially-continuous stream surface temperatures; however, it is for a single point in time (Dugdale, 2016; Torgersen et al., 2001). TIR data have been successfully used to identify spatial heterogeneity (e.g., Bingham et al., 2012) and locate groundwater and tributary inputs (Dugdale et al., 2013; Loheide and Gorelick, 2006; Mundy et al., 2017).

DTS and TIR are sometimes used in conjunction with stream temperature models. DTS data were used to calibrate and validate a 1.3 km physically-based, one-dimensional stream temperature model of the Boiron de Morges River in southwest Switzerland (Roth et al., 2010; Westhoff et al., 2007). TIR data have been used in conjunction with stationary temperature loggers to calibrate reach- and basin-scale models (Bingham et al., 2012; Cardenas et al., 2014; Carrivick et al., 2012; Deitchman and Loheide, 2012). For example, TIR data were combined with instream temperature loggers to calibrate an 86 km QUAL2Kw water quality model in the Wenatchee River in Washington (Cristea and Burges, 2009) and a 100 km scale statistical model in the Big Hole River, MT (Vatland et al. 2015). In the latter study, Vatland et al. (2015) concluded that single



point monitoring sites underestimate the temporal and spatial heterogeneity in stream temperatures and that DTS data provided a promising addition to TIR and stationary loggers. While DTS and TIR have been applied to calibrate reach-scale models, no studies have used both DTS and TIR to quantify observed temperature ranges within model reaches, and more importantly, use to this information to extend model predictions for key river features that create highly variable temperatures over small

spatial scales. Such insight to small-scale responses allows researchers, managers, and stakeholders to identify thermal micro-habitats and further interpret basin-scale model results.

       The objectives of this study were to 1) evaluate small-scale stream temperature variability, or the range of stream temperatures, using DTS data and TIR imagery, 2) use these data to corroborate an existing one-dimensional (300 m spatial resolution), basin-scale stream temperature model calibration, 3) identify river features with greater stream temperature ranges

due to spatially variable temperatures, and 4) add measured, spatially explicit stream temperature ranges to model results for appropriate river features to futher interpret temperature variability throughout a watershed. Nevada's Walker Basin was the study watershed and is representative of other arid and semi-arid watersheds in western USA where cold water species, like trout and salmon, are temperature-limited. River restoration is ongoing in the Walker Basin, focusing on environmental water purchases to improve habitat connectivity between Walker River and Lake (National Fish and Wildlife Foundation, 2011;

Walker Basin Conservancy, 2017). An existing hydrodynamic streamflow and temperature model has evaluated restoration alternatives (Elmore et al. 2015; Null et al. 2017). However, to assist in prioritizing restoration efforts, there is a clear need to understand small-scale stream temperature ranges in various river features (e.g., pools, confluences), to identify temperature barriers to migration, and to better interpret reach or watershed-scale model results.

## 2 Study Site

20       The Walker River flows from the east-slope Sierra Nevada Mountains into Walker Lake, a terminal lake in the Great Basin (Fig 1). The lower elevations of the Walker Basin have an arid climate with hot summers, whereas high elevations receive heavy snowfall during cold winters (Sharpe et. al 2008). The Walker River is a desert stream with mean annual flows of 15.5 – 30 $m^3$/s. The mainstem Walker River is the confluence of two branches, the East Walker River and the West Walker River, which flows into Walker Lake. In the prolonged drought from 2011-2017, lower portions of the Walker River were

dry, disconnecting it from Walker Lake in the fall of 2014 and 2015 (Null et al. 2017).

**Figure 1: Walker River modeled extent, June 2015 DTS deployment sites, and July 2012 TIR imagery extent.**

       Agriculture is the main land use in the basin. Irrigated farmland makes up approximately 450 $km^2$ of the 10,720 $km^2$ Walker Basin (Sharpe et. al 2008). Bridgeport Reservoir on the East Walker River, Topaz Reservoir on the West Walker, and Weber Reservoir on the mainstem Walker River regulate water to support agriculture and other human water uses. There are

23 diversions and 8 return flows in the East, West, and mainstem Walker Rivers, which influence both streamflows and stream temperatures. Interactions between climate, management actions, surface water, and groundwater are complex in the Walker Basin (Niswonger et al. 2014). The Walker River basin generally gains water during wet years and loses flow during dry





years; however, the mainstem Walker River is almost always a losing water (Carroll et al., 2010). Agricultural flood irrigation replenishes groundwater levels during the summer months (Carroll et al., 2010; Lopes and Allander, 2009).

Walker Lake once supported Lahontan cutthroat trout (LCT) (*Oncorhynchus clarkii henshawi*), which spawned in the Walker River and tributaries. The abundance and distribution of native LCT are limited by warm stream temperatures,
low streamflows, and low dissolved oxygen in the Walker River (Coffin and Cowan 1995; USFWS 2003). Historically, Walker River contained healthy populations of LCT; however, they persist in less than 10% of their historical range of the Walker River (Coffin and Cowan 1995; Dunham et al. 1999). LCT are listed as a threatened species under the Endangered Species Act (USFWS 1975). Field studies conducted in Coyote Lake, Quinn River, and Humboldt River indicate LCT occurrence is reduced at stream temperatures above the acute (< 2hr) threshold of 28 °C (Dunham et al. 2003). Measured
stream temperatures exceeded the acute temperature threshold for LCT (> 28 °C) during summer 2014 and 2015 in the Walker River (Null et al., 2017).

Low instream flows from surface water diversions have also caused Walker Lake level to decline, increasing dissolved salts in the lake to concentrations which do not support trout and native benthic insects (Herbst et al., 2013; Wurtsbaugh et al., 2017). This has largely disconnected river and lake ecosystems. To address these problems, an environmental water purchase
program acquires natural flow and storage water rights from willing sellers who switch to crops that require less water or improve agricultural water use efficiency (NFWF, 2018; Walker Basin Conservancy, 2018). To date, 2.3 m$^3$/s of natural flow water rights and 13.3 million m$^3$ of storage water rights have been purchased, approximately 40% of the water needed to restore Walker Lake (Walker Basin Conservancy, 2018). Previous research has suggested that environmental water purchases intended to increase lake elevation also improve aquatic habitat conditions in the Walker River by increasing streamflows,
reducing stream temperatures, and increasing dissolved oxygen concentrations (Elmore et al. 2015; Null et al. 2017).

## 3 Methods

DTS and TIR field measurements estimated stream temperature ranges for various river reaches and features within these reaches at small-spatial scales in the Walker River during dry years. Measured data were 1) compared to River Modeling System v4 (RMS) modeled temperature estimates to corroborate model results and 2) used to highlight river features and
locations that provided thermal barriers or refugia for native aquatic species like LCT. This section describes DTS and TIR data collection, the RMS streamflow and temperature model, statistical analyses between measured and modeled data, and methods applied to upscale model results to measured data by identifying river features like diversions, return flows, seeps, and beaver dams throughout the modeled reach.

### 3.1 Distributed Temperature Sensing (DTS) Data

DTS units measure temperatures throughout a fiber optic cable by sending a laser pulse down the cable and timing the return signal. Although the majority of energy sent into the cable is reflected back at the original wavelength, a portion of



the energy is absorbed and re-emitted at both shorter and longer wavelengths. These changes in the back-reflected wavelength frequency are Raman-backscatter. Raman-backscatter is further split into two categories, Stokes backscatter, the longer wavelength reflection, and Anti-Stokes backscatter, the shorter wavelength reflection. Temperatures along the cable are determined from the Stokes/Anti-Stokes ratio, with changing wavelength amplitude depending on temperatures (Selker et al. 2006).

A silver 1 km DTS cable was deployed to measure diurnal stream temperatures in the mainstem and East Walker Rivers. Data were collected over 400 m in the East Walker River at Rafter 7 Ranch on June 18-23, 2015 and over 450 m in the mainstem Walker River at Stanley Ranch on June 25-30, 2015 (Fig. 1). The DTS cable was deployed in a U shape at both sites, with approximately 400 m of cable on each side of the stream to capture lateral stream temperature differences. The cable was suspended in the water column approximately 10 cm above the streambed with steel stakes and leashes. Mainstem Walker River DTS deployment included approximately 20 m of a flood irrigation return flow canal named the Wabuska Drain toward the downstream end of the DTS cable. The Wabuska Drain was not flowing during the drought when the DTS was deployed, but contained standing water and was connected with the Walker River.

A two-channel Sensornet Orxy DTS unit measured stream temperatures at a spatial resolution of 1 m and temporal resolution of 15 minutes. Each data collection event measured temperatures over 30 seconds and averaged temperature along the 1 m sample interval. Measurement precision from the unit is 0.01 ºC in the -40 to 65 ºC range. The DTS had two co-located fibers within the cable that were connected in a splice box at the end of the cable. This created an internal loop of fiber, producing one double-ended set of temperature measurements (Hausner et al., 2011). However, the splice box was damaged, so two single-ended datasets were evaluated in place of one double-ended dataset. In this installation, the DTS measured temperatures on Channel 1 that covered the length of fiber from the instrument to the damaged splice; and then Channel 2 measured temperatures for the same length, on the other fiber. This resulted in two temperature measurements at each data collection point along the cable.

The DTS was dynamically calibrated during deployment with 10 m of cable placed in three recirculated calibration baths, one ambient and one ice bath near the DTS unit and one ambient bath at the end of the cable (Hausner et al., 2011; Tyler et al., 2009). RBRsolo thermocouple temperature sensors measured temperatures in calibration baths that are accurate to 0.002 ºC in the -5 ºC to 35 ºC range. Nine Maxim Integrated iButton thermistors provided additional stream temperature measurements along the cable every 15 minutes to verify DTS temperatures. iButton temperature loggers are accurate to 0.5 ºC in the -40 to 85 ºC range. Calibration compared the DTS data against thermocouple data and adjusted the DTS data to match the thermocouple data using a linear transformation. Post-collection processing used the single-ended explicit calibration method developed by Hausner et al. (2011). First, sections of cable that were exposed to air were removed from the dataset. Due to cable damage near the splice box prior to the third calibration bath, post processing relied upon iButton data closest to the end of the cable and the two calibration bath thermocouples near the DTS. Because tension on the DTS cable can result in erroneous temperature measurements (Hausner et al., 2011), data points were removed if the temperature difference between the two channels was >1 ºC. Temperature for these points were linearly interpolated between the upstream



and downstream cable locations. Root mean square errors (RMSEs) were calculated between each thermocouple or iButton and corresponding DTS temperature. We reported the average RMSE of the two thermocouples and iButton to quantify DTS error for the length of the cable for each channel. The channel with the lowest calibrated RMSE was used for data analysis and results. In addition, RMSEs were calculated between the georeferenced iButton stream temperature measurements and

the corresponding georeferenced DTS stream temperature measurements for the entire data collection period to provide additional corroboration of the DTS temperatures. iButton residuals were calculated as the difference between iButton temperatures and co-located DTS measured temperatures.

A Decagon eKo Pro Series meteorological station with an eKO ET22 weather sensor collected solar radiation, wind speed and direction, air temperature, humidity, barometric pressure, and precipitation every 15 minutes at the DTS data

collection locations for each deployment. Edge of water, DTS cable location, thalweg, and channel cross sections were surveyed with a Leica Viva GS14 GNSS Real Time Kinematic (RTK) GPS and measurements were accurate to approximately 2 cm. Survey data georeferenced the DTS cable and estimated 10 cm contours of streambed bathymetry. USGS gages 10293500 and 10301500 provided flow data for the East Walker River and mainstem Walker River, respectively. DTS deployments occurred on warm and clear summer days when maximum air temperatures were 34.7 ºC at the East Walker

River and 37.9 ºC at the mainstem Walker River DTS sites. Average flow was 1.2 m$^3$/s (42 ft$^3$/s) in the East Walker and 1.0 m$^3$/s (36 ft$^3$/s) in the mainstem Walker during deployment (Fig. S2).

### 3.2 Airborne Thermal Infrared (TIR) Data

TIR imagery of the Walker River was collected by Watershed Sciences Inc. (2012) and measured surface stream temperatures for 240 river km in the East Walker, West Walker, and mainstem Walker Rivers on July 18 and 24 - 26, 2012

(Fig. 1). A FLIR Systems, Inc. SC6000 sensor (wavelength of 8-9.2 µm, Noise Equivalent Temperature Differences of 0.035 ºC, and pixel array of 640 x 512 at a 14 bit encoding level) mounted on the underside of a Bell Jet Ranger Helicopter collected imagery, and was flown at an altitude of approximately 610 m. Pixel resolution was 0.6 m (Watershed Sciences Inc., 2012).

Watershed Sciences Inc. (2012) calibrated, georeferenced, and interpreted the TIR imagery. Surface inflow temperatures were reported at their confluence with the Walker River. Stream channel TIR temperatures were summarized by

querying temperature at ten locations in the center of the channel and reporting the minimum, median, and maximum values. We refer to these interpreted temperatures as summary points throughout this paper. Flight speed, image overlap, and river features determined which images to sample (Watershed Sciences Inc., 2012). Watershed Sciences Inc. (2012) reported stream temperature accuracy of 0.5 ºC or better for TIR imagery. TIR data were collected on warm summer days with low humidity. Average air temperature during data collection was 33.1 ºC and average wind speed was 11.6 km per hour (kph) in Yerrington,

NV. Average flow was 1.0 m$^3$/s (34 ft$^3$/s), 1.1 m$^3$/s (39 ft$^3$/s), and 2.8 m$^3$/s (100 ft$^3$/s) in the mainstem Walker River (USGS gage 10301500), West Walker River (USGS gage 10298600), and East Walker River (USGS gage 10293500), respectively (Watershed Sciences Inc. 2012). See Watershed Sciences Inc. (2012 and 2011) for additional TIR data collection detail.





### 3.3 River Modeling System (RMS) Modeled Stream Temperatures

RMS is a 1-dimensional hydrodynamic and water quality model which solves the St. Venant equations for conservation of mass and momentum and the Holly-Priessmann mass transport equation (Hauser and Schohl, 2002). Previous research provided modeled streamflows and stream temperatures for one wet (2011) and three dry (2012, 2014, 2015) irrigation

seasons (April 1-October 31) (Elmore et al. 2016; Null et al. 2017). A total of 305 km of the East Walker, West Walker, and mainstem Walker Rivers were represented in RMS at an hourly time step and 300 m modeled reach spatial resolution. Based on underlying modeling assumptions, each reach has a homogenous temperature throughout the reach. Walker River modeled extent includes the East Walker River downstream of Bridgeport Reservoir (river km 243 to 117), the West Walker River downstream of Topaz Reservoir (river km 60 to 0) and the mainstem Walker River to Walker Lake (river km 117 to 0) (Fig.

1). For additional model detail see Elmore et al. (2016) and Null et al. (2017).

### 3.4 Thermal Range and Comparison to RMS Outputs

Reach minimum, maximum, and average stream temperatures were calculated for each 15 minute DTS sample event, day, and for the entire deployment period for both DTS sites (Table 1). The 15 minute reach average temperature was calculated as the spatial average of the temperature measured at each 1 m DTS sampling point. Day and deployment period

reach average temperatures were calculated from the 15 minute spatial average following Eq. 1:

$$TSavg_{t,r} = \frac{\sum_{i=1}^{t}(Ts_{avg,i,r})}{t} \tag{1}$$

where $TSavg_{t,r}$ is the reach average temperature for time, $t$, and $TS_{avg,i,r}$ is the 15 minute spatial average for reach, $r$. Time, $t$, in Eq. 1 was day, $d$, or deployment period, $p$.

The temperature range for 15 minute DTS sample event, day, and deployment period was calculated by subtracting

the minimum measured temperature from the maximum measured temperature for the 1000 m DTS cable following Eq. 2:

$$R_{t,r} = Ts_{max,i,r} - Ts_{min,i,r} \tag{2}$$

where $R_{t,r}$ is the temperature range for time, $t$, and reach, $r$, $Ts_{max,i,r}$ is maximum measured temperature for 15 minute events, $i$, and $Ts_{min,i,r}$ is minimum measured temperature for 15 minute events, $i$, and reach, $r$. Time in Eq. 2 was day, $d$, or deployment period, $p$.

The daily and deployment period minimum, maximum, and average of reach stream temperature range were calculated from the 15 minute events for each DTS reach following Eq. 3:

$$Ravg_{t,r} = \frac{\sum_{i=1}^{t}(Ts_{max,i,r} - Ts_{min,i,r})}{t} \tag{3}$$

where $Ravg_{t,r}$ is the average reach temperature range for time, $t$, $Ts_{max,i,r}$ is maximum measured temperature for 15 minute events, $i$, and $Ts_{min,i,r}$ is minimum measured temperature for 15 minute events, $i$, and reach, $r$. Time in Eq. 3 was day, $d$, or

deployment period, $p$.





In addition, reach minimum and maximum top of the hour DTS temperatures, $Ts_{min,toh,r}$ and $Ts_{max,toh,r}$, respectively, were compared to hourly modeled Walker River stream temperatures, $T_{mod,toh,r}$, to quantify the thermal range not captured within the one-dimensional modeling. The percentage of time when modeled temperatures were outside of measured temperatures, $T_{mod,toh,r} < Ts_{min,toh,r}$ and $T_{mod,toh,r} > Ts_{max,toh,r}$, was also calculated. RMSE, mean absolute error (MAE), and mean

bias between the spatial average for the top of the hour DTS temperatures and hourly modeled temperatures summarized differences between modeled and measured data. See Wood (2017) for RMSE, MAE, and mean bias equations.

TIR temperature measurements were also compared with modeled temperatures for the Walker River from Bridgeport and Topaz Reservoirs to Weber Reservoir over the three days in July 2012 when TIR data was collected. To compare measured TIR surface temperatures with model results, TIR summary points provided by Watershed Sciences Inc. (2012) were

georeferenced with the 300 m modeled reaches. On average there were 3 TIR summary points per 300 m modeled reach. TIR flight times determined which model day and hour to compare with TIR temperatures. The spatial average of minimum, maximum, and average TIR temperature for each 300 m modeled reach was calculated for the East Walker, West Walker, and mainstem Walker Rivers following Eq. 4:

$$TSavg_L = \frac{\sum_{j=1}^{L}(Ts_{avg,r})}{L} \tag{4}$$

where $TSavg_L$ is the East, West, or mainstem Walker River, $L$, spatially averaged temperature and $Ts_{avg,r}$ is the mean of summary point median stream temperatures for modeled reach, $r$, (i.e., the average 300 m modeled reach temperature) because TIR summary points reported minimum, maximum, and median temperatures only. River length, $L$, in Eq. 4 was the East Walker, West Walker, and mainstem Walker Rivers.

The spatial average of the TIR stream temperature range within each 300 m modeled reach for the East Walker, West

Walker, and mainstem Walker Rivers was calculated following Eq. 5:

$$Ravg_L = \frac{\sum_{j=1}^{L}(Ts_{max,r} - Ts_{min,r})}{L} \tag{5}$$

where $Ravg_L$ is the river, $L$, spatial average temperature, $Ts_{max,r}$ is the maximum summary point temperature for the 300 m modeled reach, $r$, and $Ts_{min,r}$ is the minimum summary point temperature for the 300 m modeled reach, $r$.

RMSE, MAE, and mean bias were calculated between the average 300 m modeled reach temperature, $Ts_{avg,r}$, and the

corresponding modeled temperature to quantify differences. The percentage of time when modeled temperatures, $T_{mod,toh,r}$, were outside of measured temperatures, $T_{mod,toh,r} < Ts_{min,r}$ and $T_{mod,toh,r} > Ts_{max,r}$, was calculated.

**Table 1: Description of stream temperature variables.**

To extrapolate model outputs that provide a uniform estimate of stream temperature for each 300 m modeled reach, measured DTS and TIR temperature ranges at river features like agricultural return flows, diversions, beaver dams, and seeps

were applied to the model outputs. This provided an estimate of spatial variability missing in the model output that is needed to gain insight into potential habitat availability at smaller-spatial scales. However, the key features must be identified throughout the area of interest to determine where to apply the appropriate variability. Diversion and return flow locations



were developed in 2012 by the Walker Basin Project (Tim Minor, pers.comm, 2012). Seeps were identified during TIR surveys. We used seep locations identified during the winter TIR flight completed on November 16-17, 2011 because temperature differences were larger and thus more obvious than the summer flight (Watershed Sciences Inc., 2011); however, we applied the temperature range observed at seeps during the summer 2012 TIR flight (Watershed Sciences Inc. 2012).

5         Beaver are native to the Walker Basin (Gibson and Olden, 2014) and beaver dams were identified using 2012 and 2013 Google Earth aerial imagery (Google Earth Pro, 2018). Locations were georeferenced where beaver dams were seen spanning the channel. Often turbulence was observed below the dam and sometimes crowdsourced photos added images of the beaver dams from the ground. We relied primarily on 2012 imagery, unless it was unavailable or of poor quality, when 2013 aerial imagery was used. 2012 and 2013 were dry years, and beaver dams were more abundant in the Walker River 10  during dry years, when high flow events that limit beavers ability to dam across the stream channel were reduced (Nevada Department of Wildlife, 2016).

## 4 Results

### 4.1 DTS Measured Stream Temperatures and Ranges

        Temperature differences were observed between DTS channels, potentially from stress on the cable. Channel 2 had 15  the lowest RMSE values, with average RMSE between calibrated DTS data and the three reference temperatures of 0.09 $^{\circ}$C and 0.15 $^{\circ}$C at the East Walker River and mainstem Walker River DTS sites, respectively (Table S1). iButton stream temperature measurements provided an additional test of DTS measurements. Average DTS error for both sites was within the 0.5 $^{\circ}$C precision of the iButtons. iButton residuals vs. DTS temperatures showed that iButtons measured warmer temperatures than the DTS for the East Walker River, although the average bias for all iButtons was within the 0.5 $^{\circ}$C precision 20  of the iButtons. There were no significant residual trends in errors for the mainstem Walker River (Table S2 and Fig. S1).

        Overall, average DTS stream temperatures in the East Walker River were approximately 4 $^{\circ}$C cooler and less variable than the mainstem Walker River (Fig. 2 and Table 2). Stream temperatures varied spatially throughout the mainstem Walker River, which are apparent as longitudinal color striations at different locations longitudinally in Figure 2. The East Walker River data show consistent colors longitudinally, indicating more consistent temperatures through the length of the reach for 25  sampled time periods. Average DTS temperature ranges within model reaches for the deployment were nearly 2 $^{\circ}$C greater in the mainstem Walker River than the East Walker River. The East Walker River DTS site is farther upstream and close to Bridgeport Reservoir, a bottom release dam. The mainstem Walker River DTS site is 92 km downstream from the East Walker River DTS site and also receives contributions from the West Walker River, fed by surface water releases from Topaz Reservoir. These results confirm those of other studies showing that stream temperatures warm longitudinally during summer 30  (Elmore et al. 2015).

**Figure 2: Stream temperatures measured for the length of the DTS cable at East Walker River (a) and mainstem Walker River (b) DTS sites. Wabuska Drain is located at cable distance 110-175 m in the mainstem Walker River site (b).**



**Table 2: Daily stream temperatures and ranges for DTS deployment reaches in the East Walker and mainstem Walker Rivers. Data collection began in the afternoon on deployment days, June 19th and 25th, and ended in the morning of June 23rd and 30th.**

In the East Walker River, deployment period minimum stream temperature was 16.7 ℃ and occurred between 6:15 and 8:30 am. Deployment period maximum temperature was 24.9 ℃ and occurred between 5:00 and 5:30 pm. Daily maximum temperatures were measured in a straight, homogenous, unshaded section (Fig. 3a). The daily minimum reach temperature range occurred between midnight and 8:15 am, while daily maximum reach temperature range occurred between 1:00 and 3:00 pm (Table 2). Reach stream temperature range of 15 minute collection events extended from a minimum of 0.5 ℃ to a maximum of 2.0 ℃ for the deployment period, with an average of 1.0 ℃. A shaded backwater eddy and pools with overhanging shrubs and tall cottonwoods were river features with increased thermal heterogeneity in the East Walker River (Fig. 3a).

**Figure 3: East Walker River daily maximum stream temperatures on June 21, 2015 at 5:30 pm (a) and 15 minute temperature range during DTS deployment (b). Modeled reach points represent the division between 300 m modeled reaches.**

In the mainstem Walker River, average reach temperature for 6/25/15 – 6/30/15 was 25.2 ℃, not including the Wabuska Drain segment (Table 2, excluding distance 110 – 175 m in Fig. 2b). Deployment maximum stream temperature was 32.9 ℃ and daily maximum stream temperatures occurred between 2:15 and 4:30 pm. The daily maximum reach temperature range occurred between 2:00 to 3:45 pm, roughly the same time as daily maximum stream temperatures were observed. The average reach temperature range for the deployment was 2.7 ℃, with a minimum reach temperature range of 1.1 ℃ and a maximum reach temperature range of 7.0 ℃. Daily minimum reach temperature ranges occurred around 9:30 am.

When the 20 m section of the Wabuska Drain return flow canal (shown approximately at distance 110 – 175 m in Fig. 2b) was analyzed with the mainstem Walker River, daily minimum and maximum temperatures changed little because they occurred in the mainstem Walker River. However, the maximum 15 minute reach temperature range for the deployment increased considerably from 7.0 ℃ to 10.2 ℃ and average reach temperature range for the deployment also increased from 2.7 ℃ to 3.6 ℃ (Table 2, Fig. 2b). Temperature range increases with the inclusion of the Wabuska Drain because it contained cooler water during hot times of the day, providing a lower minimum temperature than observed in the mainstem Walker River, while also containing warmer temperatures near the shallow outlet. Because daily minimum and maximum temperatures did not change, but the reach temperature range increased with the inclusion of the Wabuska Drain, the Wabuska Drain likely receives cool groundwater inputs which pool in the canal without lateral mixing with warmer water in the mainstem river (Fig. 2b and Fig. 4a). This allows Wabuska Drain to provide a cool water refuge from the hot temperatures in the mainstem during the day when agricultural return flows are limited and water in the drain is likely dominated by groundwater inflow. The cool water preserved in Wabuska Drain increased reach temperature range during hot times of the day, driving the increase in observed daily maximum reach temperature range and daily average reach temperature range.

**Figure 4: Mainstem Walker River daily maximum stream temperature on June 29, 2015 at 3:15 pm (a) and 15 minute temperature range during DTS deployment (Wabuska Drain temperatures are not included) (b). Model reach points represent the division between 300 m model reaches.**



Figure 4 illustrates the cooling effect of the Wabuska Drain and the spatial temperature variability during daily maximum stream temperatures on July 29th at 3:15 pm. The coolest temperature in the mainstem Walker River DTS site was 24.4 °C and occurred approximately 20 m into Wabuska Drain (Fig. 4a). Warm stream temperatures of up to 31.8 °C occurred in the homogeneous mainstem Walker River segment just upstream of the Wabuska Drain outlet along the shallow, right bank.

While the Wabuska Drain provided an overall cooling effect on the mainstem Walker River, it was a river feature with increased thermal variability with warm stream temperatures at the mouth and cooler stream temperatures within the Wabuska Drain. The shallow Wabuska Drain also experienced rapid heating and cooling in response to atmospheric conditions. In addition, cool water from the outlet of Wabuska Drain mixed with the mainstem Walker River, expanding the temperature range of the downstream segment as well. Stream temperatures in the shallow water at the mouth of Wabuska Drain and in

the mainstem Walker River upstream of the Wabuska Drain exceeded LCT acute temperature threshold of 28 °C and thus may be thermal barriers to fish passage during summer afternoons.

In addition to increased temperature ranges in the Wabuska Drain, the mainstem Walker River had more channel and temperature heterogeneity from inactive, breached beaver dams. On June 29th at 3:15 pm, when reach average temperature was 29.6 °C, nearly 7 °C of the temperature range observed for this 15 minute sample event occurred at a breached beaver dam

(Fig. 4a). The warmer temperatures occurred in an unshaded, shallow, backwater location subject to increased solar heating. Cooler temperatures occurred in the pool created by the dam, making the deeper pool a potential temperature refuge for fish.

### 4.2 TIR Measured Stream Temperatures and Range

While DTS measurements provided high spatial and temporal stream temperature resolution at two sites, TIR measurements provided improved spatial resolution at one hour for surface stream temperatures in the Walker River. TIR

temperatures showed a general longitudinal warming trend, with stream temperatures increasing 9 °C from Bridgeport Dam to Weber Reservoir. Consistent with model results (Elmore et al. 2015), the coolest observed temperature, 20.1 °C, occurred in the East Walker River and the hottest observed temperature, 29.2 °C, occurred in the mainstem Walker River (Table 3). The average TIR temperature range within 300 m modeled reaches was 0.3 °C within the East Walker River, 0.4 °C within the West Walker River, and 0.3 °C within the mainstem Walker River (Table 3).

**Table 3: Stream temperatures and temperature ranges within 300 m modeled reaches by river from July 2012 TIR remotely-sensed data.**

Maximum stream temperatures typically occurred in reaches with canal diversions and return flows. Maximum reach temperature in the East Walker River of 26.5 °C (Table 3) occurred at the Hall Diversion (River km 129). The maximum stream temperature of 27.1 °C in the West Walker occurred upstream of the confluence with the mainstem Walker River.

Maximum temperature in the mainstem Walker River of 29.2 °C occurred in the reach immediately downstream of the Wabuska Drain return flow (River km 78), which may create temperature barriers for cold water species like LCT at some times. Although Wabuska Drain was receiving agricultural returns and therefore contributing warm water, rather than the cool water observed during times with limited agricultural returns, cooling of 1 °C over 4.5 river km was observed downstream of



the Wabuska Drain. This response may be due to additional increased groundwater inflows downstream of the Wabuska Drain consistent with valley narrowing (Watershed Sciences Inc., 2012). While increased groundwater influence may be less obvious when the return canal is flowing, the DTS results showed evidence of cool water inputs when the canal was not flowing. Thus, large diversions and return flows can create warm water conditions when active, but they also recharge shallow aquifers that

5  can increase groundwater contributions and create pockets of cold water.

300 m reaches with the greatest temperature range corresponded to locations with canal diversions, return flows, and groundwater seeps (Fig. 5a). TIR results at the basin-scale support DTS findings of increased temperature range at Wabuska Drain measured at the Stanley Ranch DTS deployment site (River km 78) (Fig. 5). 300 m reach temperature range was also larger in the East Walker River at the Fox/Mickey Diversion (River km 126) and Strosnider Diversion (River km 140) (Fig.

5b). In the mainstem Walker River, there was more thermal variability at the Spragg-Alcorn-Bewely Diversion (River km 94), the Spragg-Alcorn-Bewely Canal Return (River km 90), and Wabuska Drain (River km 78) (Fig. 5). TIR summary reports did not include the locations of beaver dams, and TIR surface temperatures are unable to capture thermal stratification of beaver dams and ponds. The maximum 300 m reach temperature range was 1.2 ºC in the West Walker River (River km 58), which did not correspond to a diversion, canal return flow, or beaver dam, but is the location of a groundwater seep (Watershed

Sciences Inc., 2012). Thus, large diversions and return flows alter river depth and thermal mass while seeps increase temperature ranges by creating a relatively consistent cool water location.

**Figure 5: Temperature range within each 300 m model reach (a) and summarized by river section (b, c, and d) from July 2012 TIR remotely-sensed data with the upstream-most river km on the left side of the x-axis.**

**4.3 RMS Predictions vs. Measured Temperatures**

Unsurprisingly, thermal variability, measured as temperature range, varied at spatial scales smaller than the 300 m model reaches used in RMS. The model did not capture the maximum and minimum temperatures within modeled reaches measured by both the DTS and TIR approaches (Fig. 6). However, it is important to note that some of this error is due to measurement errors within the DTS and TIR observations. In particular, TIR data capture surface water temperatures, which may overestimate water column temperatures from vertical stratification and thermal boundary layer effects (Torgerson et al.

2001) and DTS data in shallow areas may be influenced by solar radiation penetration in the water column (Neilson et al. 2010).

**Figure 6: Top of the hour DTS minimum and maximum temperatures compared to model predictions in the East Walker River (a) and mainstem Walker River (b) DTS sites (Wabuska Drain temperatures are not included as they were not modeled). July 2012 TIR minimum and maximum temperatures compared to modeled temperatures for East Walker (c), West Walker (d), and mainstem**
**Walker (e) Rivers. The upstream end of Weber Reservoir is at river km 48. The upstream most river km is on the left side of the x-axis in panels c - e.**

Predicted temperatures in 2015 were greater than DTS maximum top of the hour temperatures 50% of the time in the East Walker River, and 20% of the time in the mainstem Walker River. Conversely, the model under predicted temperatures 29% and 10% of the time in the East Walker and mainstem Walker Rivers, respectively. Temperatures measured in Wabuska

Drain were excluded from this analysis because the model estimated temperatures in the main channel only. The RMSE for





the DTS deployment length was 1.1 ºC in the East Walker River and 1.7 ºC in the mainstem Walker River (Table 4). Interestingly, the model over estimated temperatures in the East Walker River with a bias of 0.2 ºC for the deployment period, but under estimated temperatures in the mainstem Walker River with a bias of -0.4 ºC for the deployment period (Table 4).

**Table 4: RMSE, MAE, mean bias, and percent of modeled dataset outside of measured values for the East, West, and mainstem Walker Rivers between modeled and top of the hour DTS and TIR stream temperatures.**

Predicted temperatures were lower than TIR minimum temperatures within a 300 m modeled reach for 74%, 95%, and 87% of survey extent in the East Walker, West Walker, and mainstem Walker Rivers, respectively (Table 4). In addition, predicted temperatures were greater than the TIR maximum temperatures within a 300 m modeled reach for 9%, 0%, and 8% of survey extent in the East Walker, West Walker, and mainstem Walker Rivers, respectively. The RMSE and bias were both <1 ºC for the East and West Walker Rivers; however, the RMSE in the mainstem Walker River was 3.4 ºC and bias was -2.5 ºC (Table 4) where the model performed poorly under low flow conditions (Fig 6e).

RMS temperature predictions were within the DTS measured temperature ranges 21% and 70% of the time for the East Walker River and mainstem Walker River, respectively. Predicted temperatures were within TIR measured temperatures 17%, 5%, and 5% of the time for the East Walker, West Walker, and mainstem Walker Rivers, respectively (Table 4). However, mainstem Walker River TIR stream temperatures vs. modeled stream temperature is the only RMSE value that exceeded the calibrated RMS model RMSE (Table 5). Model bias for the East Walker River indicated the model over estimated stream temperature by 0.2 ºC in the 300 m DTS reach over the five day study period and underestimated temperature by 0.5 ºC for the 77 km TIR extent. The model underestimated stream temperatures by 0.4 ºC from the average DTS values measured at the top of the hour and underestimated stream temperatures by 2.5 ºC when compared to the TIR average temperature in the mainstem Walker River (Table 4).

## 5 Discussion

### 5.1 Limitations

Obtaining small-scale spatial and temporal stream temperatures and comparing them to model results has a number of limitations. DTS data quality can be impacted by instrument drift during multi-day deployments and drift can be as large as 1-2 ºC from cable stress and rapid fluctuations in internal DTS temperature (Tyler et al., 2009). Field crews monitored DTS cables daily and it was assumed that the DTS cable did not move during deployments. Deployments used leashes to hold the cable in place and minimize cable stress as much as possible; however, evidence of minimal cable stress was observed from different temperatures of the two DTS channels (Table S1). In addition, solar heating of the DTS cable was assumed to be negligible because the cable was silver coated to reflect solar radiation (Tyler et al., 2009) and solar heating of DTS cables would be limited in advection-dominated and turbid rivers (Neilson et al., 2010), such as the Walker River. Low RMSE with iButton data supports this assumption.



Surface roughness, surface emissivity, surface reflection, variable background temperatures (e.g., sky versus trees), turbidity, changes in viewing aspect, aircraft type, flight speed and wind gusts all affect TIR image and data quality (Dugdale, 2016). The length of time required to collect TIR imagery can also impact the quality of data because stream temperatures change during the course of the survey (Dugdale, 2016); this was minimized in this study by collecting all TIR imagery on warm, clear days with similar weather conditions. Unless noted, we assumed a vertically mixed water column when analyzing the DTS and TIR data, which is again reasonable for advection-dominated streams. However, the skin of the water surface can be a different temperature than the water column, potentially creating a source of error in TIR data (Torgersen et al., 2001). Additionally, pools and beaver dams may stratify vertically, increasing the local temperature variability from what was measured or predicted.

## 5.2 Walker River Habitat Implications from DTS and TIR Stream Temperature Measurements

Warm stream temperatures and low flows threaten native trout and other cold water species in the Walker River. This research measured small-scale thermal variability, or range of stream temperatures, that was unquantified and underrepresented in existing basin-scale modeling. The East Walker River did not exceed the acute 28 ℃ temperature threshold for LCT and had less variable stream temperatures because of its location high in the watershed and is located downstream of a bottom release reservoir. Maximum DTS temperatures in the mainstem Walker River were 4.5 ℃ warmer than the acute temperature threshold of 28 ℃ for LCT (Dunham et al. 2003); however, the Wabuska Drain provided cold water refuge during warm summer days when agricultural return flows were not present. The cold water in Wabuska Drain decreased 15 minute reach minimum temperatures during warm times of the day, increasing the maximum DTS deployment temperature range by 3.1 ℃ and average DTS deployment temperature range by 0.8 ℃ in the mainstem Walker River. Lopes and Allander (2009) identified local streamflow gains near the Wabuska gage, hypothesizing they originated from groundwater inputs to Wabuska Drain. However, shallow subsurface water, or interflow contributions to Wabuska Drain may not occur during the entire year, particularly outside of irrigation season as groundwater levels decline or with groundwater depletion from drought conditions (Naranjo and Smith, 2016).

The greatest temperature range in the Walker River DTS study reaches occurred in the early afternoon of summer days and at canal diversions, return flows, beaver ponds, and backwater eddies. Beaver dams had high spatial and temporal temperature ranges, consistent with findings from Majerova et al. (2015) and Weber et al. (2017). A 15-minute temperature range of 7 ℃ was observed in a beaver dam in the mainstem Walker River. Cristea and Burges (2009) observed 2 - 3 ℃ temperature range due to cold water seeps or channel braiding in the Pacific Northwest, which is comparable to the 1 – 2 ℃ temperature range observed in the East Walker River in the DTS data and TIR imagery. Return flow channels, beaver dams, and seeps likely create thermal refugia during some time periods, improving aquatic habitat connectivity between Walker Lake and River for cold water, migratory species.





### 5.3 One-Dimensional Model Result Interpretation

To provide greater insight into watershed-scale responses, measured DTS and TIR temperature ranges from return flows, diversions, beaver dams, and seeps were applied to one-dimensional stream temperature predictions to identify potential micro-habitats, temperature barriers, and temperature refugia in the basin. Overall, we identified 23 diversions, 8 return flows,

53 possible seeps, and 42 beaver dams throughout the modeled reach of the West Walker, East Walker, and mainstem Walker Rivers (Fig 7a). Observed temperature ranges varied around modeled temperatures from -10.1 to 2.3 °C for return flows, -1.2 to 4 °C for diversions, -5.1 to 2 °C for beaver dams, -4.2 to 0 °C for seeps. Average temperature change from nearby stream conditions was -2.5 °C for return flows, +1.2 °C for diversions, -3.2 °C for beaver dams, and -1.9 °C for groundwater seeps. Applying observed temperature ranges from DTS and TIR observations suggests that cool-water refugia exist to support

species migration between Walker Lake and upper tributaries of the Walker River (Fig 7b).

**Figure 7: Locations of river features that affect stream temperatures in the Walker Basin (a). Predicted RMS stream temperatures with average temperature change and estimated ranges by river feature using DTS data from June 29, 2015 at 3:15 pm and TIR data from July 18 and 24 - 26, 2012) (b).**

Previous research has shown that the Walker River has poor aquatic habitat as a function of streamflow, stream

temperature, dissolved oxygen concentrations, food abundance, and substrate from the confluence of the East and West Walker Rivers to Walker Lake, but that the East and West Walker Rivers are still likely to support native aquatic species  (Elmore et al., 2015; Hogle et al., 2014; Mehler et al., 2015; Null et al., 2017). Research suggests that water purchases and other restoration actions that prioritize passage through the lower Walker River to re-connect river and lake ecosystems are likely to be more effective than restoring suitable habitat in the lower Walker River (Hogle et al., 2014; Null et al., 2016, 2017).

Environmental water purchases, the primary restoration action in the Walker Basin, can effectively reduce daily maximum temperatures during periods of low flow by increasing thermal mass (Elmore et al. 2015). Environmental water purchases may be most beneficial for stream temperatures in mid to late afternoon, the time when maximum temperatures occur in the mainstem and East Walker Rivers. Results show specific river features like diversions, return flows, beaver dams, and large eddies provide small-scale temperature variability. Cold-water refugia potentially allow trout populations to persist

where surrounding stream temperatures exceed thermal tolerance limits (Sutton et al., 2007). However, trout use of thermal refugia may vary, as availability of refugia change with streamflow and weather conditions, and as trout habitat needs vary with life stage (Dugdale et al. 2013). Future research is needed to validate temperature ranges by river feature at the watershed-scale, evaluate how fish use thermal refugia, and to improve understanding of the resiliency of thermal refugia with anticipated climate change.

Augmenting environmental water purchases with secondary restoration efforts at canal return flows and beaver dams could further preserve cold water observed in both DTS and TIR datasets. Secondary restoration efforts should focus on minimizing thermal barriers and enhancing cold water refugia to improve habitat connectivity and mitigate warm stream temperatures in the Walker River. Results identified warm water segments that may act as thermal barriers to fish passage in shallow, unshaded reaches at the mouth of irrigation diversions and return flow outlets, stagnant edges of beaver dam pools,



and in homogenous, unshaded habitat segments. Promising secondary restoration efforts include native riparian vegetation restoration to reduce heating due to solar radiation, creating channel complexity to increase habitat quality, and increasing thermal variability by re-introducing beaver, designing beaver dam analogs restoration efforts, or adding large wood to the river (Bond et al., 2015; Poole and Berman, 2001; Weber et al., 2017). While restoration is ongoing to preserve the riparian

corridor and promote native habitat by reducing grazing and removing invasive plants (USFWS, 2017), other secondary restoration projects depend on the extent to which stakeholders want to manage habitat and restoration.

## 5.4 Complementing Process-based Modeling with DTS and TIR Measurements

DTS and TIR data provide insight into the variability of stream temperatures that are not captured by individual sensors located in the channel. We show the utility of these data for one-dimensional model validation, but also extend the use

of these data to further quantify the possible spatial variability occurring within model reaches containing key features that create thermal heterogeneity not captured by simplistic models. In other words, measurements can be used to complement process-based modeling. Results contribute to literature describing thermal refugia networks and how they may be considered and maintained with reservoir releases, riparian restoration, or other river restoration approaches (Isaak et al., 2010; Seavy et al., 2009; Sutton et al., 2007). Coupling high resolution stream temperature monitoring with process-based modeling can make

model simulations more useful for identifying temperature barriers, refuges, and promising restoration strategies. It provides a more realistic stream temperature range than one-dimensional modeling alone, especially when model results assess habitat suitability or evaluate watershed-scale river management and restoration alternatives. It also may be widely applied by stakeholders who do not have the funding or background to conduct additional model simulations, but prefer to post-process results with measurement data.

This is the first study using both DTS and TIR to quantify small-scale temperature range within one-dimensional stream temperature model reaches. Using DTS and TIR to compare stream temperature measurements to predicted stream temperatures helps to bound spatial temperature range and can be applied in other watersheds to identify habitat features that are important for understanding small-scale temperature ranges and restoration management. This research uses the Walker River to demonstrate how an increased understanding of the temperature ranges present within modeled reaches can be used

to interpret model results, supplying vital information for restoration decision makers.

## 6 Summary

Small-scale (micro-habitat) stream temperature ranges and timing were measured using DTS and TIR. Observations were coupled with an existing one-dimensional (300 m resolution) stream temperature model to identify temperature barriers and refuges at the watershed-scale. Stream restoration that maximizes cold water refugia mitigates warm stream temperatures

to increase habitat quality and connectivity for native fishes. Understanding small-scale temperature ranges is useful to re-interpret watershed-scale stream temperature results and identify river features that provide thermal refugia or thermal barriers




to migration.  Beaver dams and return flow channels maximize temperature ranges and may mitigate warm stream temperatures.  Restoration should maintain and enhance these features to improve aquatic habitat connectivity.

*Acknowledgements:*

This research was funded by the National Fish and Wildlife Foundation (grant number 2010-0059-201).  The DTS
was provided by the Center for Transformative Environmental Monitoring Programs (CTEMPs), funded by the National Science Foundation (award EAR 0930061).  Thank you to Scott Tyler and Scott Kobs at the University of Nevada, Reno and Mark Hausner at the Desert Research Institute for DTS expertise and guidance.  Thank you also to Nathaniel Mouzon, Kelley Sterle, Zack Arno, Hannah Friedrich, and Curtis Gray for field assistance, and to Brett Roper for feedback on an early version of this paper.



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



**Table 1: Description of stream temperature variables.**

| Variable | Metric | Temporal Extent | Spatial Extent |
|---|---|---|---|
| | | DTS | |
| $Ts_{min,t,r}$ | Minimum Temperature | 15 minute (i), Top of Hour (TOH), Day (d), Deployment Period (p) | 300m |
| $Ts_{max,t,r}$ | Maximum Temperature | | |
| $Ts_{avg,t,r}$ | Average Temperature | | |
| $R_{i,r}$ | Temperature Range $Ts_{max,i,r}$ -$Ts_{min,i,r}$ | 15 minute | |
| $R_{min,t}$ | Minimum of $R_{i,r}$ | | |
| $R_{max,t}$ | Minimum of $R_{i,r}$ | Day (d), Deployment Period (p) | |
| $R_{avg,t}$ | Average of $R_{i,r}$ | | |
| $T_{mod,toh,r}$ | Modeled Stream Temperatures | Top of Hour | |
| | | TIR | |
| $Ts_{min,r}$ | Minimum of Summary Points | Hour of Flight Collection | 300 m |
| $Ts_{max,r}$ | Maximum of Summary Points | | |
| $Ts_{avg,r}$ | Average of Summary Point Medians | | |
| $R_r$ | Temperature Range $Ts_{max,r}$ -$Ts_{min,r}$ | | |
| $Ts_{min,L}$ | Minimum of $Ts_{min,r}$ | | East, West, or mainstem Walker River |
| $Ts_{max,L}$ | Maximum of $Ts_{max,r}$ | | |
| $Ts_{avg,L}$ | Average $Ts_{avg,r}$ | | |
| $R_{max,L}$ | Maximum of $R_r$ | | |
| $R_{avg,L}$ | Average of $R_r$ | | |
| $T_{mod,toh,r}$ | Modeled Stream Temperatures | Top of Hour | 300 m |





**Table 2: Daily stream temperatures and ranges for DTS deployment reaches in the East Walker and mainstem Walker Rivers. Data was only collected in the afternoon on deployment days, June 19th and 25th, and only in the morning of demobilization days, June 23rd and 30th.**

| | Minimum | | | | Maximum | | | | Average | |
|---|---|---|---|---|---|---|---|---|---|---|
| | Min. Temp. (ºC) | Min. Temp. Time | Min. Range (ºC) | Min. Range Time | Max. Temp. (ºC) | Max. Temp. Time | Max. Range (ºC) | Max. Range Time | Avg. Temp (ºC) | Avg. Range (ºC) |
| East Walker River | | | | | | | | | | |
| 6/19/15 | 19.8 | 11:15 | 0.6 | 19:45 | 24.9 | 17:00 | 1.4 | 13:00 | 23.1 | 1.0 |
| 6/20/15 | 18.0 | 6:15 | 0.5 | 8:30 | 24.9 | 17:30 | 2.0 | 13:00 | 21.3 | 1.1 |
| 6/21/15 | 18.0 | 6:15 | 0.5 | 23:30 | 24.4 | 17:30 | 1.5 | 13:45 | 21.2 | 0.9 |
| 6/22/15 | 16.7 | 8:30 | 0.5 | 0:30 | 24.0 | 17:30 | 1.7 | 14:45 | 20.3 | 1.0 |
| 6/23/15 | 17.3 | 8:00 | 0.5 | 8:15 | 21.0 | 0:15 | 1.1 | 9:45 | 18.9 | 0.7 |
| **Overall** | **16.7** | **8:30** | **0.5** | **8:15** | **24.9** | **17:00** | **2.0** | **13:00** | **21.0** | **1.0** |
| Mainstem Walker River including Wabuska Drain | | | | | | | | | | |
| 6/25/15 | 22.0 | 14:15 | 3.6 | 23:45 | 32.9 | 16:15 | 10.2 | 16:00 | 28.6 | 7.1 |
| 6/26/15 | 21.0 | 6:30 | 1.6 | 23:00 | 29.9 | 14:15 | 6.5 | 14:15 | 25.0 | 3.8 |
| 6/27/15 | 21.8 | 7:00 | 1.4 | 9:15 | 31.0 | 15:45 | 6.7 | 15:45 | 25.8 | 3.0 |
| 6/28/15 | 21.8 | 8:00 | 1.4 | 9:30 | 26.9 | 16:30 | 3.2 | 16:30 | 24.3 | 2.2 |
| 6/29/15 | 21.0 | 6:00 | 2.0 | 8:30 | 31.9 | 15:15 | 7.5 | 15:15 | 25.2 | 3.7 |
| 6/30/15 | 20.0 | 6:45 | 2.4 | 10:00 | 29.5 | 12:30 | 6.3 | 12:30 | 23.1 | 3.5 |
| **Overall** | **20.0** | **6:45** | **1.4** | **9:30** | **32.9** | **16:15** | **10.2** | **16:00** | **25.2** | **3.6** |
| Mainstem Walker River excluding Wabuska Drain | | | | | | | | | | |
| 6/25/15 | 23.7 | 23:45 | 2.2 | 19:15 | 32.5 | 16:15 | 7.0 | 15:30 | 28.8 | 3.9 |
| 6/26/15 | 20.0 | 6:30 | 1.2 | 21:00 | 29.9 | 14:15 | 4.5 | 14:00 | 25.1 | 2.5 |
| 6/27/15 | 21.8 | 7:00 | 1.1 | 9:30 | 31.0 | 15:45 | 3.4 | 15:45 | 25.8 | 1.8 |
| 6/28/15 | 21.8 | 8:00 | 1.2 | 9:30 | 26.9 | 16:30 | 3.1 | 15:45 | 24.4 | 2.0 |
| 6/29/15 | 21.0 | 6:00 | 1.8 | 9:45 | 31.9 | 15:15 | 7.0 | 14:00 | 25.3 | 3.5 |
| 6/30/15 | 20.0 | 6:45 | 2.3 | 10:00 | 29.5 | 12:30 | 5.7 | 12:30 | 23.1 | 3.4 |
| **Overall** | **20.0** | **6:45** | **1.1** | **9:30** | **32.5** | **16:15** | **7.0** | **15:30** | **25.2** | **2.7** |

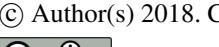



**Table 3: Stream temperatures and temperature range within 300 m modeled reaches by river from July 2012 TIR remotely-sensed data.**

|  | Minimum Temperature (ºC) | Maximum Temperature (ºC) | Average Temperature (ºC) | Maximum Range (ºC) | Average Range (ºC) |
|---|---|---|---|---|---|
| East Walker River | 20.1 | 26.5 | 24.7 | 1.1 | 0.3 |
| West Walker River | 24.1 | 27.1 | 25.6 | 1.2 | 0.4 |
| Mainstem Walker River | 22.9 | 29.2 | 27.3 | 1.0 | 0.3 |

**Table 4: RMSE, MAE, mean bias, and percent of modeled dataset outside of measured values for the East, West, and mainstem Walker Rivers between modeled and top of the hour DTS and TIR stream temperatures.**

|  | RMSE (ºC) | MAE (ºC) | Mod. – Meas. Bias (ºC) | Mod. > Meas. (%) | Mod. < Meas. (%) | n (hrs) |
|---|---|---|---|---|---|---|
| East Walker River DTS | 1.1 | 0.9 | 0.2 | 50 | 29 | 94 |
| mainstem Walker River DTS | 1.7 | 1.3 | -0.4 | 20 | 10 | 118 |
| East Walker River TIR | 0.8 | 0.6 | -0.5 | 9 | 74 | 2 |
| West Walker River TIR | 0.9 | 0.8 | -0.8 | 0 | 95 | 1 |
| mainstem Walker River TIR | 3.4 | 2.7 | -2.5 | 8 | 87 | 3 |
| Walker River Overall TIR | 1.9 | 1.2 | -1.1 | 7 | 83 | 6 |

Table 5: Comparison of stream temperature RMSE between RMS model vs. DTS data, RMS model vs. TIR data, and reported RMS model fit

|  | RMSE (DTS vs. RMS model) (ºC) | RMSE (TIR vs RMS model) (ºC) | RMSE (Reported RMS model fit) (ºC) |
|---|---|---|---|
| East Walker River - 2015 | 1.1 | -- | 2.1[a] |
| mainstem Walker River - 2015 | 1.7 | -- | 1.8[a] |
| East Walker River - 2012 | -- | 0.8 | 2.0[b] |
| West Walker River - 2012 | -- | 0.9 | 1.4[b] |
| mainstem Walker River - 2012 | -- | 3.4 | 2.5[b] |
| Walker River Overall - 2012 | -- | 1.9 | 2.1[b] |

[a] Null et al. 2016, [b] Elmore et al. 2016





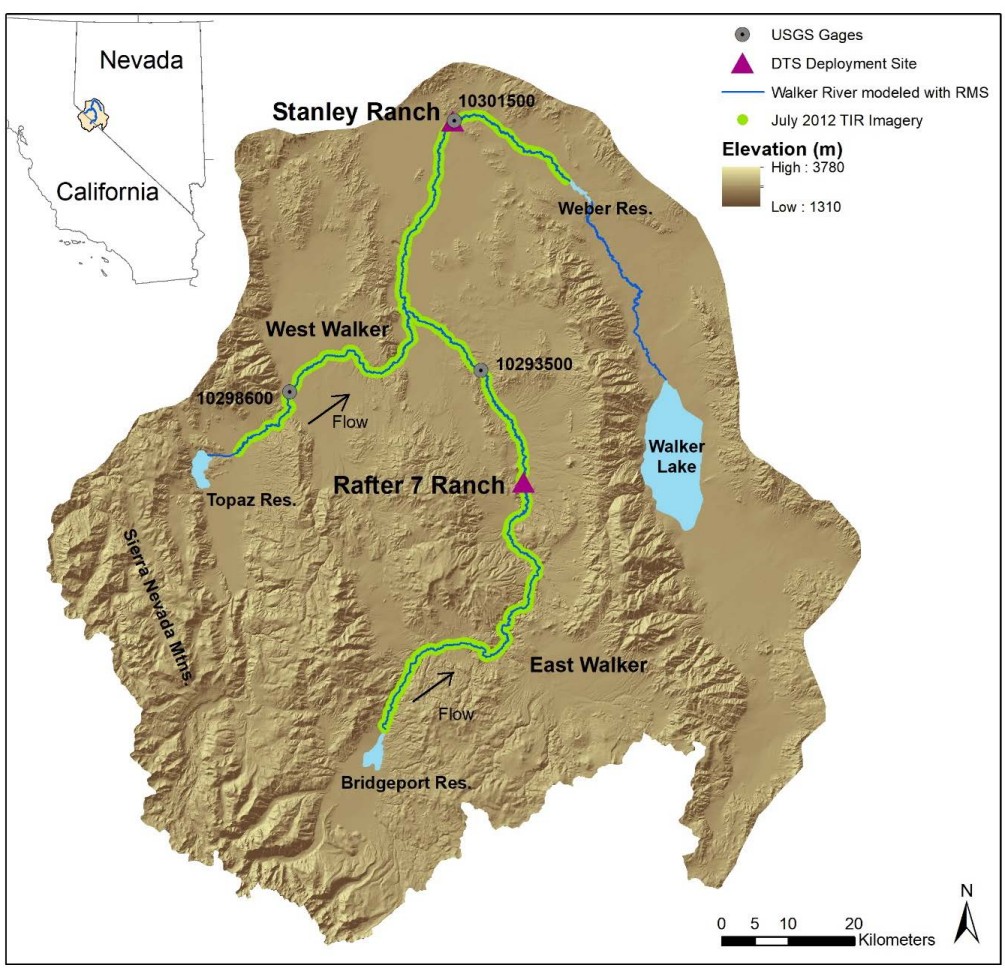

**Figure 1**



(a)

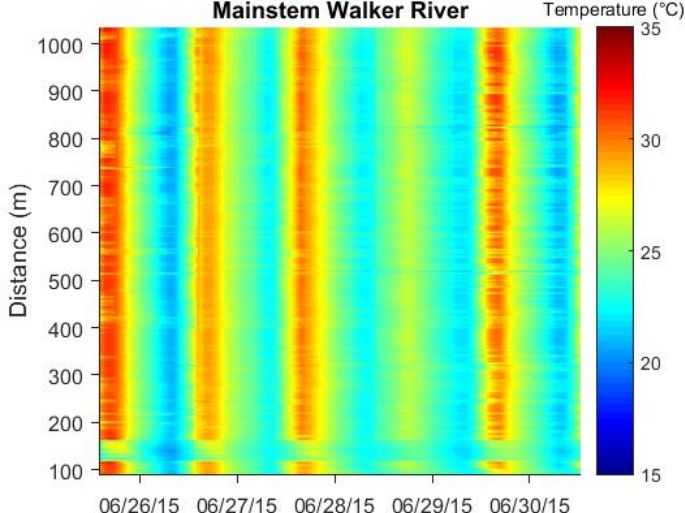

(b)

**Figure 2**




(a)

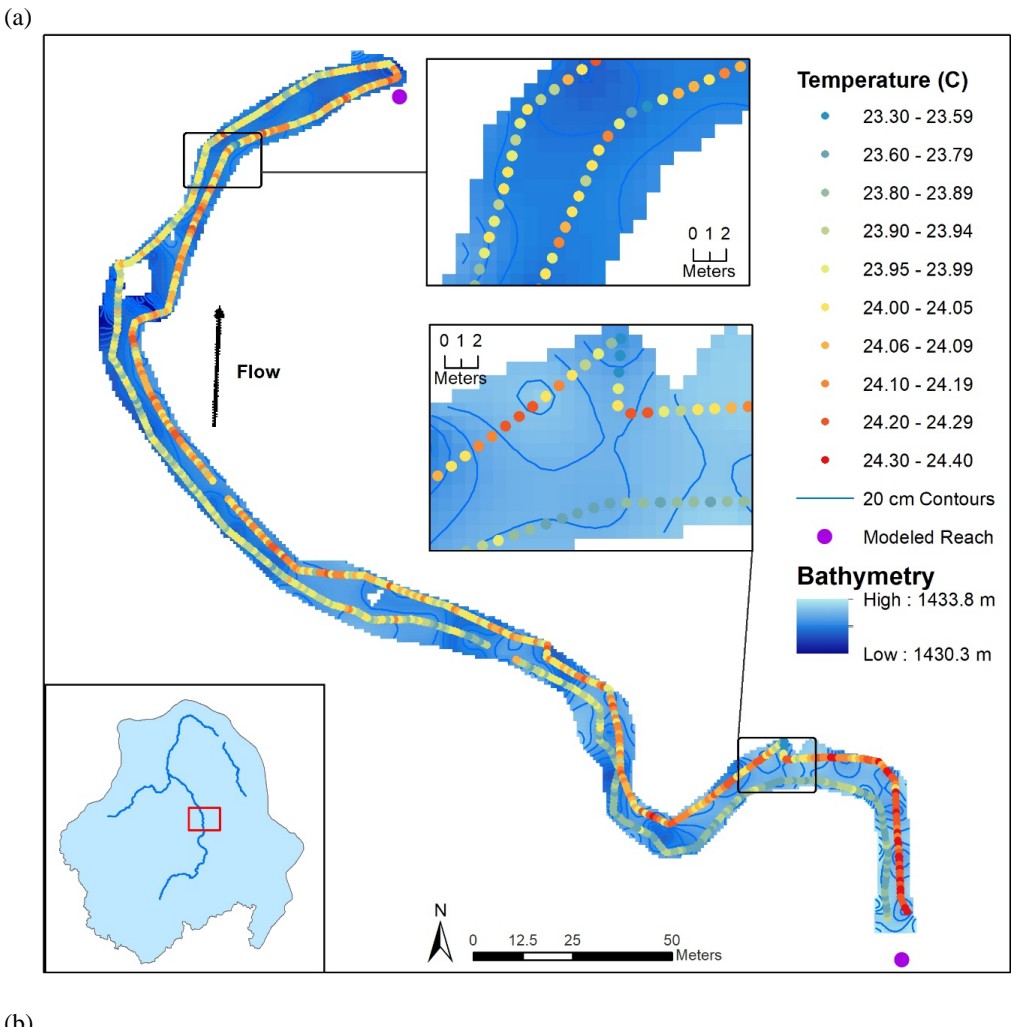

(b)

**Figure 3**





(a)

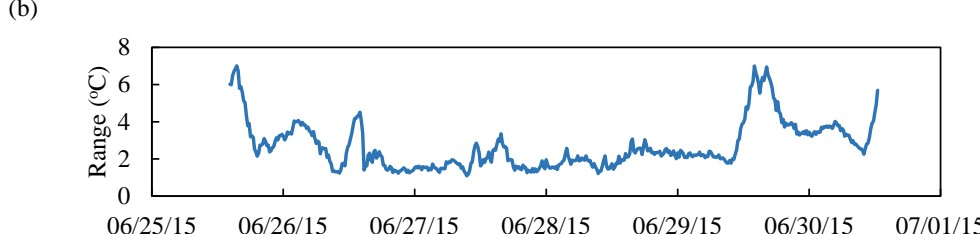

**Figure 4**



(a)



**Figure 5**





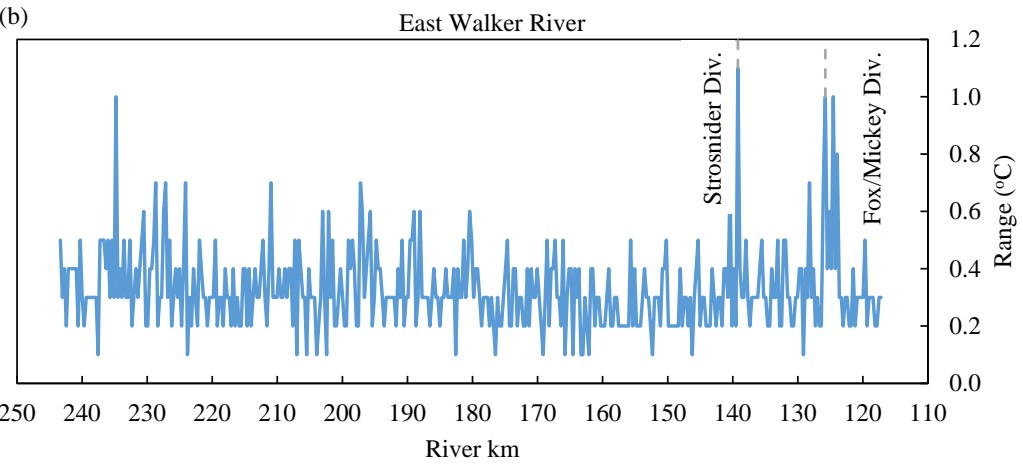

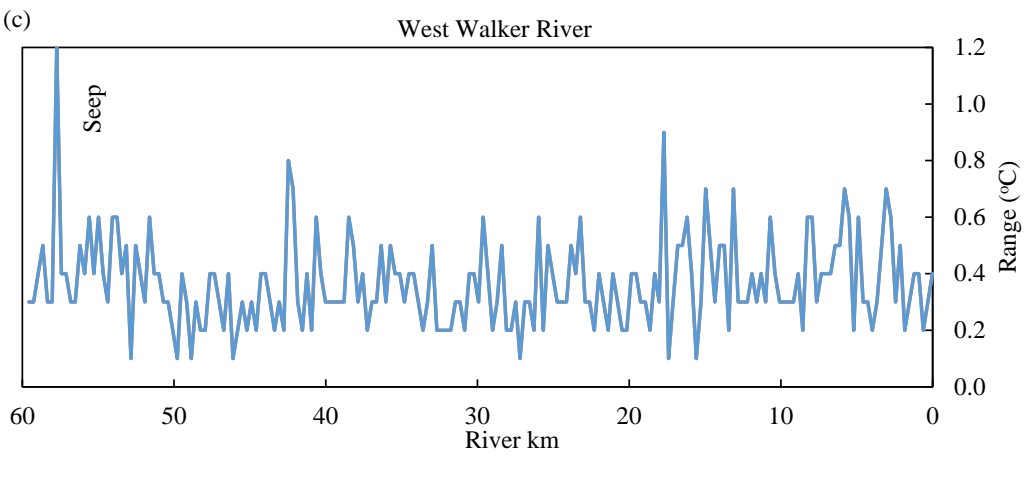

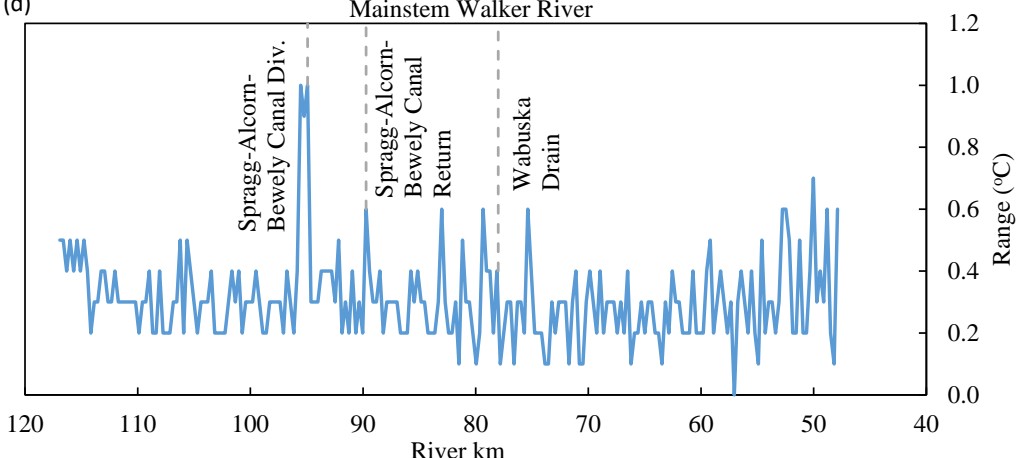

**Figure 5**





(a)

(b)

(c)

(d)

(e)

Modeled — Minimum — Maximum

**Figure 6**



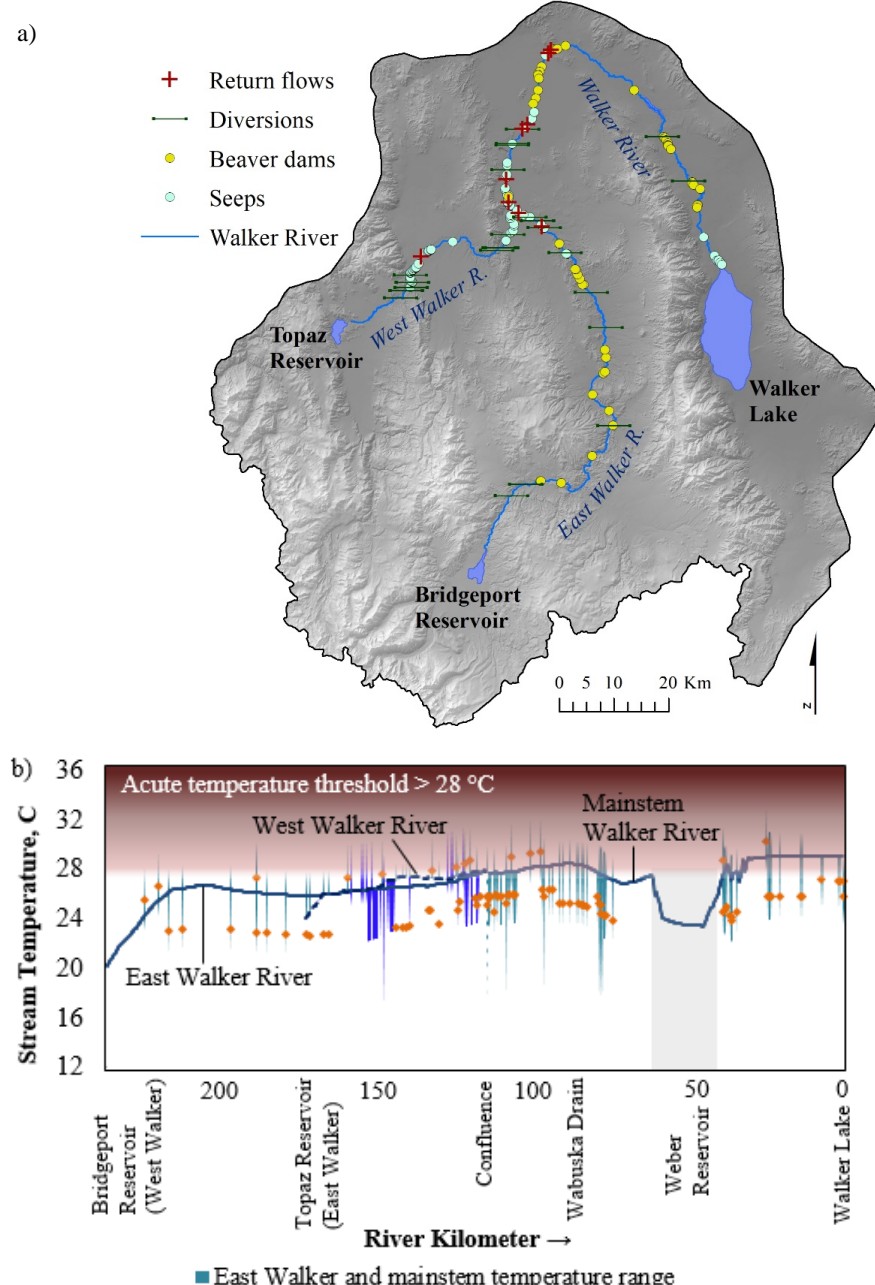

**Figure 7**