# Peer review of "Quantifying Thermal Refugia Connectivity by Combining Temperature Modeling, Distributed Temperature Sensing and Thermal Infrared Imaging"

_Hydrology and Earth System Sciences, 2018_

## Referee Comment (RC1) · Anonymous Referee #1 · 17 Oct 2018

General Comments:

The paper addresses relevant scientific questions focusing on the importance of spatial and temporal fine scale variations in thermal habitat for Pacific salmon. The introduction could focus more on setting up the problem of warming waters for Pacific salmon in the southern extent of their range and on the importance of small scale variability in temperature for fish survival. Examining more recent papers that reference Sutton et al 2007 could help in this effort. Combining DTS, TIR, and one-dimensional temperature modeling in an effort to better manage and restore river systems is unique.

The methods were outlined reasonably well; however there were some points of confusion. It is possible that a cartoon schematic of the deployment of the DTS could be beneficial. The DTS itself had "two channels" and at times the wording became confusing because the DTS was deployed in a river channel. Results largely consisted of summary statistics of the temperature data and modeled temperature. It would be instructive to do more analysis that relates to the Pacific salmon of interest and its 28 degrees C threshold. Additional investigation might be done to research the following questions: 1)What percentage of the deployment was 28 degrees C exceeded at the mainstem site? 2) When the temperature was exceeded, did the model over or underestimate the actual value and by what percentage? 3) Is there anyway to go beyond the summary statistics of the dataset? The conclusions reached are clear; however, they seem fairly straightforward in that a one-dimensional model would certainly miss fine scale variation that occurs in a real river channel. I think it could be shown more clearly and in further detail the times when the one-dimensional model succeeds or fails for the Pacific salmon. How certain is that the results presented would not be different for a deployment time later in the summer? How might the results and discussion change for a period later in the summer? Key numerical results could be added to the abstract. At times the writing does not appear to have a singular voice.

Specific Questions:

pg. 3, Line 30: In what way do the diversions and return flows influence the temperature? It could be beneficial to explicitly state they warm the waters.

pg. 4, Line 10: What percentage of the deployment period did temperatures exceed 28 degrees C? Is this a common problem in the watershed (are there locations where it is always greater than 28 degrees C) or does the problem occur only during discrete heatwave events?

pg. 4, Line 30: Is it necessary to describe how DTS works in this level of detail?

pg. 8, Line 28: How were key features added to the model output?

pg. 9, Line 29: It would be useful to show that there was not a large warming even that occurred between the two deployments at the two sites. Weather conditions should be ruled out as a factor of influence on the two deployments.

Specific Notes: pg. 2, Line 3: Therefore, assessing stream temperatures...

pg. 2, Line 9: Stream temperature models are a useful tool for river management because they help...

pg. 2, Line 23: ...spatially-continuous stream surface temperatures. However, it... What does "it" refer to?

pg. 3, Line 31: Interactions between... "between" compares two things while "among" compares more than two things

pg. 4, Line 12: Low instream flows from surface water diversions have also caused the Walker Lake level to decline

pg. 5, Line 25: Verb tense was changed to present tense.

---

## Referee Comment (RC2) · M.C. Westhoff (Referee) · 17 Oct 2018

In this manuscript, high spatial resolution temperature observations are used to investigate small scale spatial variability in stream temperatures. These are placed on top of simulated stream temperatures with grid cells of 300m, leading to the conclusion that at moments when the simulated temperature exceeds the critical temperature (for LTC) of 28 C, there are always pockets of water that are cool enough to serve as refuges.

Although these findings are potential useful, this manuscripts reads a bit as an engineering report with the focus on 'improving' an existing 1D 300m-resolution temperature model. There is little in-depth science involved: the results section is mainly a summing up of observations at the 300 m scale, while more quantitative assessment of the uncertainties/sensitivities of the used approach is missing.

Because I still think the available data is sufficient to improve the manuscript significantly, I am advising major revisions.

What in my opinion is needed is a more in-depth analysis about spatial variability over a whole range of scales. For example: one can compare the left and right bank DTS measurements for each cross section as the smallest scale ($\sim$1m). Subsequently, the temperature range for progressively larger scales can be obtained. Finally, the implication for the model scale (300m) can be used as a practical case study. Similar things can be done for the TIR observations. Although it was not clear to me if the 10 points which were used to obtain the summary points were measured in 1 cross section or if they were measured over 300m, with a spatial resolution of 0.6 m, there seems to be enough data to do the same analysis for the TIR data.

Line by line comments:

P2,L29: Westhoff et al. (2007) did their research in the Maisbich river in Luxembourg.

P3, L3: Neilson et al. (2010) did indeed not use these measurements, but they did model the temperature of transient storage zones.

P3,L10: isn't spatial variability not the same as temperature range?

P3, L23: please also indicate stream width and depth estimates.

P4, L31: change to: the majority of the reflected energy has it original wavelength.

P6, L25: Are these 10 locations spread over 300 m, or 10 locations covering one cross-section? And why are only values in the centre of the channel taken? Pools etc, are generally located on the sides of the channel. The spatial variability will probably be

larger when these values on the side of the channel will be taken into account as well.

Section 3.4: Here, many statistical parameters are explained. As a reader I found it difficult to remember what the meaning is of each statistic. Furthermore, the symbols defined here hardly ever comes back in the results section. My suggestion here is to only define a few of them, and in the results section clearly state if a spatial or temporal range is discussed.

P7, L16: use subscripts for TSavg: now it reads as T*S*a*v*g (the same for all other equations). Instead of 'avg' one can also use a bar on top of T

P8, L1: what does "top of hour" mean?

P8, L10: If I understood it correctly, these summary points were the mean, min and max. So why are there "on average" 3 summary points? This implies that at some location you have less and at others you have more summary points.

P8, L15: L is river length, isn't it?

P8, L28: What do you mean with "extrapolate model outputs"?

P8, L31: the key features are the same as the river features discussed in the previous lines, aren't they?

P11, L15: I guess you mean that the backwater location is subject to a higher thermal mass (or lower heat capacity)?

P11, L33: with a single snapshot in time one cannot conclude that the stream is cooling. One can only see that it is cooler downstream, but this can also be caused by warm water that only travels slowly downstream.

Section 5: Discussion: I miss a discussion on the difference between TIR and DTS measured ranges: which one has the largest range and highest temperature, and why? Although both methods measured at different periods in time, it must be possible to say something about it.

P14, L4-5: Why is the change in temperature over time minimized on warm days? I would presume that on such days the rate of change is at its maximum, since also the daily amplitude is the largest on these days

P14, L8-9: Can you quantify the effect of stratification?

P15, L6: The definition of temperature range is defined as Tmax-Tmin, so it can never be a negative value.

Section 5.4: this belongs in the conclusion (or summary) section.

Figures, general comment: add the caption of the figures also right below the figures: I know this will happen once the manuscript is published, but as a reviewer, I found it annoying that these captions were only listed in the text.

Fig. 3b: reduce the scale on the vertical axis to max 2 C?

Fig.5: In my opinion, there is little added value to the panels b-d. I also wouldn't call them "summaries" anyway: it is just a different way of plotting the same results.

Fig.7: What was the time for the simulated temperature? I am also wondering how variable the range over time is. This is especially interesting since the TIR data is from a completely different year than the DTS measurements. The legend of panel b can also be improved: What is the dark blue line, how are the average temperatures obtained?

References:

Westhoff, M. C., Savenije, H. H. G., Luxemburg, W. M. J. ., Stelling, G. S., van de Giesen, N. C., Selker, J. S., Pfister, L. and Uhlenbrook, S.: A distributed stream temperature model using high resolution temperature observations, Hydrol. Earth Syst. Sci., 11(4), 1469–1480, doi:10.5194/hess-11-1469-2007, 2007.

Neilson, B. T., S. C. Chapra, D. K. Stevens, and C. Bandaragoda (2010), Two‐-zone transient storage modeling using temperature and solute data with multiobjective calibration: 1. Temperature, Water Resour. Res., 46, W12520, doi:10.1029/2009WR008756.

With kind regards,

Martijn westhoff
* * *

---

## Referee Comment (RC3) · Anonymous Referee #3 · 22 Oct 2018

Hydrology and Earth System Sciences manuscript HESS-2018-441 describes the result of a study to quantify small-scale temperature variability in the Walker River, Nevada, using a combined approach leveraging DTS, thermal infrared (TIR) remote sensing and one-dimensional river temperature modelling. DTS and TIR data were incorporated with coarser temperature model predictions to identify potential thermal refugia habitat that may otherwise be missed from lower-resolution temperature model outputs. While the general idea of combining TIR, DTS and model predictions is laudable (and there is definitely a need for this type of research), I felt that the manuscript

was generally too descriptive, with a lack of formal hypothesis testing or deeper analysis. Furthermore, the method by which the TIR/DTS data was combined with the temperature model predictions (through simple addition of temperature ranges to the temperature model-derived thermal long profiles) was rather simplistic and does not appear to offer a substantial benefit over the use of the DTS/TIR data alone. Despite these issues, I support the general concept of the manuscript and believe that the idea of combining DTS, TIR and model predictions has merit, but I feel that significant revisions (including a much more in-depth treatment/analysis of the data beyond the scope of what would normally be considered 'major revisions') is needed prior to publication.

—General comments—

-In section 3.4 (and table 1), it is quite difficult to remember/follow the names of the different variables. Is there any way of streamlining/simplifying this?

-It's probably not necessary to give all of the detail about the DTS system (ie. 2 channels, cable stress, etc) in the methods and results sections. Consider removing all superfluous information to streamline the manuscript.

-It would be nice to see some of the TIR imagery to allow for comparison with the DTS data; given the large time difference between the TIR and DTS data, the information in table 3 is of limited use.

-Use of the word 'RMS' (ie. the name of the model) is confusing in places when you are also talking about RMSE. Consider putting 'RMS' in italics or similar to avoid confusion.

-There is very little information about the RMS model in terms of inputs, implementation, etc. I appreciate that this information is already given in the other references provided, but it would nonetheless be useful for see it outlined in this MS (maybe a short paragraph explaining model function, input data, etc).

-It's not immediately clear in the manuscript how you 'apply' the DTS/TIR temperature data to the model. From my reading, you simply add/subtract the temperature range

to the model outputs. Is this correct? If so, I would have thought that this process (ie. calculating temperature variability from DTS and TIR and simply adding/subtracting to/from the model outputs) is accompanied by a range of issues given that the DTS data is essentially a measure of spatio-temporal temperature range, whereas the TIR data only gives spatial variability. Also, given that the TIR data and DTS data were acquired at different times, it does not seem appropriate to overlay these data onto the RMS model output for the same point in time (eg. fig 7).

-What is the reason for the large temperature discrepancy between the two river banks? This kind of information is quite interesting and could benefit from a thorough treatment in the manuscript.

-Section 3.4 of the methodology is difficult to follow. Given that this section contains the majority of the analyses covered in the results and discussion sections, it would be beneficial to restructure this section to make the subsequent results and discussion easier to follow.

—Introduction—

P2 L1: Would be good to have 1-2 sentences of more general information on the importance of river temperature in the context of climate change.

P2 L24: It would be a good idea here to qualify the point about TIR only providing a single snapshot in time by saying something along the lines of 'unless acquired multiple occasions'.

—Methods—

P4 L22-28: These lines are potentially redundant and could be moved or redistributed elsewhere in the methods section.

P4 L30-31 and P5 L1-5: I'm not sure that a description of how DTS works is necessary; consider removing this paragraph (or at least, shortening it to one sentence)

P5 L6-13: Consider adding a figure to illustrate the deployment of the DTS. I appreciate that this is partially covered in fig 1 (large scale) and fig 3/4, but it would be nice to see a full resolution map showing the DTS installation.

P5 L18-22: Is this information necessary? Consider removing.

P5 L28: This sentence formulation is slightly difficult to follow. Do you mean to say that the calibration process consisted of using a linear transform to correct the DTS based on the difference between the DTS and thermocouple temperatures in the ice bath?

P6 L18-32: There isn't any mention here of the winter TIR data collection flights which you subsequently refer to later on in the manuscript. The manuscript only appears to have dates and hydrometeorology information for the summer flights. I appreciate that the winter data is only used for locating seeps, but it would be good to talk about the winter flights here first.

P6 L32: Although I managed to find the Watershed Sciences documents online, they were quite difficult to track down. It would therefore be good to give some further brief details of the flights, for example RMSE or R2 of TIR data vs. logger values, etc.

P7 L7: Sentence ('each reach [. . .] throughout the reach') is a little clunky – consider rewriting.

P8 L28: This sentence ('To extrapolate model outputs. . .') is difficult to follow; it is difficult to understand exactly what you are doing here. Can you think of a different way to explain this step in the methodology?

P8 L32 and P9 L1: What do you mean by 'were developed in 2012'? Do you mean that diversions/return flows were implemented in 2012 in the model, or in reality? Or just that they were mapped/identified?

P8 L2-4: If you wanted to exhaustively identify all seeps, I wonder if a better practise would have been to combine data from the winter and summer survey flights? Also, please give some detail about how the seeps were identified. Was it from manual photo

interpretation? Were aerial photos acquired simultaneously to aid the interpretation process?

—Results—

P9 L26-30: These sentences would be better suited to the discussion section.

P10 L8-9: I'm assuming that the 'shaded backwater eddy' and 'pools with overhanging shrubs' are the inset panels in fig 3(a)? If so, please label these panels in the figure, as it is not clear what they are.

P10 L20: I'm not sure what you mean here when you say that the daily max/minimum temperature changed little when analysed with the Walker River. Do you mean to say that a lack of large-scale temperature variability in the study reach masks considerable localised variability in areas like the Wabuska drain?

P10 L25-30: Some of this material might be better suited to the discussion section.

P11 L32-33 and P12 L1-5: This would also be more suitable to the discussion.

—Discussion— P13 L23-31 and P14 L1-9: I'm not really sure what information this 'limitations' section adds to the manuscript. There are clearly limitations when conducting an approach such as this combining TIR, DTS and modelling. However, this section reads more like a list of problems associated with TIR and DTS data collection (which have already been well established in the literature) rather than a critical appraisal of the inherent difficulties of combining and comparing these types of data with 1D temperature models, which would be much more interesting (and potentially useful for the reader).

—Figures—

Figure 7(b). As discussed above, it is not clear how you combine the RMS stream temperature data with DTS data from June 2015 and TIR data from July 2012. Surely this mixing up of different dates and times means that the temperature ranges from the

DTS cannot be comparable to those from the TIR? Also, to what does the 'average temperature' refer? Is it from the DTS data (temporal) or is it average temperature (spatial) calculated as the mean of all pixels covering the refugia/beaver dam, etc?

---

## Referee Comment (RC4) · Anonymous Referee #4 · 2 Nov 2018

General Comments:

The article compares temperature data from DTS and thermal imagery within a stream reach where one-dimensional temperature modelling was conducted. The study takes place in the Walker Basin, NV, where stream temperatures can exceed thermal tolerance for native, threatened trout. This study provides a unique combination of methods and is useful in understanding the pros and cons of each. The study is relevant for work that seeks to restore habitat for fish in streams where temperatures rise above the thermal tolerance of threatened cold-water species.

[Figure]

There are a few places where the manuscript could be improved. The authors generally summarize the differences between the methods, but they could take it to the next level with doing a more quantitative analysis comparing the different methods. At the end, there is not a strong conclusion, or list of pros and cons, comparing DTS vs TIR methods for validating or contributing to stream modelling efforts. What would the authors decide to use - DTS or TIR - based on the results of this paper?

Second this paper has the data to conduct analyses that look at the availability of different water temperatures that are relevant for trout. For example, the authors focus on 28C as a thermal cut off for LCT, but these fish may become thermally stressed at much lower temperatures (e.g., 21 or 22C). Additionally, very cold temperatures in the summer may not be ideal for growth. The authors could consider adding an analysis that describes the area of stream that is available to LCT below different cut offs, e.g., below 18C, below 21C, below 25C, and below 28C, and the distance between those areas. The DTS and TIR methods identify river features that the one dimensional model doesn't, but how important/large/cold is the water provided by these features? The authors could also compare how the different methods perform in quantifying the amount and connectivity of thermal refugia.

One way that could improve the flow of the results section would be to put the comparison of all three methods up front, and then describe the results of how temperatures differ in the basin second. These are currently split into different sections, with the temperatures differences in the basin described following each method. Additionally, the description of the DTS channels can be a bit confusing because there are river channels and DTS channels – is there another descriptor that could be used in place? The focus on the Wabuska diversion can also be confusing, because it was flowing during the time window of one method and not flowing for the time window of the other method.

Finally, the introduction and discussion could use more context for why micro thermal regufia is important for cold water fish and trout. The author should look for more recent

citations of Sutton et al. 2007, Brewitt and Danner 2014, etc.

Specific Comments:

Pg 2 Line 5. It could be useful to explain here explicitly what you mean by one dimension. (Obvious to stream temp. modelers but less intuitive to managers). Do you mean longitudinal interpolations along the stream?

Pg 2 Line 13-15. Similar comment as above – what does 2, vs 3 dimensions mean for streams? Length, width, depth, and in that order?

Pg 2 Lines 23-25. Switch the order of these sentences. Describe how TIR has been used to locate various stream features, but then the downside is that it is a single snapshot in time.

Pg 3 Lines 2-5: This sentence is key in setting up what your study contributes to the ones that you cite, but it isn't clear. These methods have been used to calibrate reach scale models but hasn't been used to quantify temperature ranges within model reaches? The difference in the wording is very slight, and needs to be clarified.

Pg 3 Lines 9 – 10 Objective #3 is has circular wording, it's not clear what the objective is. Is the goal to dentify features with greater temperature ranges because they have variable temperatures? Maybe the objective is just to identify those features?

Pg 3, Lines 11-18 I suggest reducing this description of the Walker Basin to 1 sentence, maybe 2 sentences here. You get into more detailed description in the next section. For example, environmental water purchases seem relatively uncommon, and you do them more justice in the next section.

Pg 4 Second paragraph – It would be nice to include some description about how narrow the range of LCT is. It makes your study more special.

Pg 4 Line 8 – Are all these study sites in CA? For a European journal, the places should be better located.

[Figure]

Pg 4 Line 14 - This sentence is a little unclear as to what it means in context of the previous sentence. Because the lake is inhabitable, it means the lake and stream systems are disconnected?

Pg 4 Line 17 – Restore to tolerable salinity ranges? Restore is a broad word. The discussion of salinity is really interesting, but perhaps not relevant to your stream temperature focus.

Pg 4 Line 22 – Can you be more specific about what a 'dry year' means in this basin? I.e., received <25% of historical rainfall, or stream base flows were at XX% of the average flows for that time of year? (Especially since you do have USGS gauge data)

Pg 4 Line 26 – Comparing measured and modeled data? (rather than between)

Pg 6 Line 31 – the East Walker has more flow than the mainstem – presumably because of diversions? Remind the reader of that again

Pg 9 Line 30 – was the Elmore et al. 2015 study also conducted in the Walker R basin? This point may be better situated in the discussion

Pg 9 – Fig 2 caption – A reminder of what is the Wabuska Drain (not-flowing, but standing water from an ag ditch) would help the reader in the caption

Pg 10 – Fig 3 – The sub-panels in Fig 3 should be labelled. In looking at the figure, it's not clear why these two features are called out. It looks like there is more variation between river left and river right than anything else. Fig 3 b could probably be moved to supplemental.

Pg 10 – Why do Fig 3 and Fig 4 visualize different times, 5:30 pm vs 3:15 pm?

Pg 10 Lines 19-31 – How different would this look if the Wabuska drain was flowing? Is it more often flowing or not flowing in the summer months?

Pg 10 – Fig 4 – It would be nice to add a line or circle indicating where the beaver dam is on that reach of stream. Fig 4 b could also be moved to supplemental. However it

might be worth exploring why there is day-to-day variation in the temperature ranges – could plot the data against air temperature range for the day?

Pg 11 Lines 10 – 11 Have LCT been observed using the Wabuska drain when it is flowing, or when stream temps aren't so high at the mouth?

Pg 11 Lines 31-32 Are there studies that show that LCT cannot move through warm temperatures? Maybe save the temperature implications for fish for the discussion, when you can do a more thorough review of studies on LCT (and related salmonids) and their thermal tolerance. They may be able to move through a small area of warm temperatures. What may be more important is the extent of cold water refuges, which could be quantified with the given data.

Pg 11 – Lines 33-34 – Make it clear that the cool water from Wabuska was observed with the other method/time window when stream temps were monitored.

Pg 12 – Fig 5 – Similarly to the other figures, the sub-panels b,c,d could be moved to supplemental since they are redundant with the main figure. Alternatively, they may be a better way to present the data in the main figure, so you could consider dropping panel a.

Pg 13 – Lines 16-19 – Model estimated within 1C – that's pretty good! But under estimating by 2.5C is less desirable – could be a point to discuss.

Pg 14 – Lines 13-15, Maybe it didn't exceed 28C, but trout can be thermally stressed at much lower temperatures.

Pg 14 Line 23 – Is this study from the Walker R Basin, or generally citing groundwater depletion from drought?

Pg 14 Line 30 – How migratory are LCT? Would they historically migrate between lakes and streams, and was that during the summer? If not, then you could shift your focus to movement and opportunities for longitudinal connectivity rather than migration.

Pg 15 Lines 9-10 Describing that there is a range in temperatures doesn't necessarily mean that there is thermal refugia. How cold is the water around these features? If it has high variability but it still warm, then it may not provide refuge for LCT.

Pg 15 Lines 27-29 This would be a good place to give a nod to the work that has evaluated how fish use thermal refugia (do a forward search on Sutton et al 2007).

---

## Author Comment (AC1) · 7 Jan 2019

**Response to Reviewers**

Thank you to the reviewers and editor for careful review of our paper "Quantifying Small-scale Temperature Variability using Distributed Temperature Sensing and Thermal Infrared Imaging to Inform River Restoration".  We are confident that it will improve the paper.

Reviewers unanimously recommended further analysis for our paper.  Following the HESS interactive review process, in which we publish a response to review comments and the editor reviews then invites a revised manuscript or rejects the paper (step 5 here: https://www.hydrology-and-earth-system-sciences.net/peer_review/interactive_review_process.html), we organize this response into

1) a description of additional analyses that we will complete if invited to submit a revised manuscript, and
2) detailed responses to all other reviewer comments that we have addressed in the manuscript because it was easier for us to make changes as we responded to reviewer comments. Reviewers' recommendations are in black and our responses are in blue.

**Additional Analyses of DTS and TIR Temperature Variability**

All reviewers have responded positively to using distributed temperature sensing (DTS) and thermal infrared (TIR) sampling to understand spatial stream temperature variability over a range of scales and comparing those data with modeled stream temperatures to identify possible locations and extents of thermal refugia for Lahontan cutthroat trout (LCT), noting that this is a unique and needed line of research.  However, all reviewers recommended further analyses to move beyond summary statistics. We agree with the reviewers that additional analyses are possible with our existing data and that they will significantly improve the paper so that it is advances understanding of thermal refugia connectivity for cold-water fish species.  If we are invited to submit a revised version of the manuscript, we will add the following analyses:

1) Quantify the percentage of DTS and TIR temperatures that exceed 21 °C (temperatures preferred by adult LCT), 24 °C (chronic 7-day upper thermal limit), and 28 °C (lethal threshold for LCT) in space and time.  We will include the temperatures and/or percentages by which the simulation model under-or over-estimates measured temperatures to better understand the times and locations that 1D modeling succeeds of fails for trout.  This will also allow for a direct comparison of DTS and TIR for quantifying the amount and connectivity of cool-water refugia (recommended by Reviewers 1 and 4).
2) Compare left and right streambank DTS measurements.  This will evaluate lateral temperature variability at a 1 m scale.  We will then evaluate temperature ranges at different spatial scales (e.g., 1 m, 10 m, 100 m, 300 m) and discuss implications to interpret 1-D model results (recommended by Reviewers 2 & 4).
3) Similarly, we will compare TIR measurements at various scales (e.g., lateral pixels and longitudinal pixels of increasingly longer reaches) to complete a similar evaluate of spatial scale with TIR data.  We have the raw TIR data which we will use rather than the summary points from the Watershed Sciences (2012) analysis (Reviewer 2).
4) We will add the TIR imagery with a DTS deployment (e.g., overlay the data or show side by side panels to Figure 3 and/or 4 to compare DTS and TIR data and results for quantifying the amount and connectivity of thermal refugia (Reviewer 3).

These analyses will advance our work beyond summary statistics to evaluate temperature ranges and alternative spatial scales so that we can better compare sampling and modeling methods and discuss the pros and cons of each method.

**Authors Response to Reviewer 1**

General Comments:

The paper addresses relevant scientific questions focusing on the importance of spatial and temporal fine scale variations in thermal habitat for Pacific salmon. The introduction could focus more on setting up the problem of warming waters for Pacific salmon in the southern extent of their range and on the importance of small scale variability in temperature for fish survival. Examining more recent papers that reference Sutton et al 2007 could help in this effort. Combining DTS, TIR, and one-dimensional temperature modeling in an effort to better manage and restore river systems is unique.

We added the following text and citations to the first paragraph in the introduction: "*Trout and salmon avoid heat stress by sheltering in pockets of cold water when stream temperatures are near upper thermal tolerances (Dunham et al. 2003; Sutton et al. 2007) and climate change is anticipated to increase stream temperatures in summer, fall, and winter (Isaak et al. 2012). Recent research has quantified when and where cold-water fish need thermal refugia (Brewitt and Danner 2014), estimated the size of thermal refugia and the distance between refugia (Fullerton et al. 2018), demonstrated how fish use thermal refugia (Frechette et al. 2018), and measured the length of time that fish can survive between refugia (Pepino et al. 2015). Where stream temperatures are warming or where cold-water fish species are at the southern extent of their range, assessing stream temperatures at small temporal and spatial scales is thus important to quantify stream temperature heterogeneity and best manage complex and variable riverine habitats (Vatland et al. 2015).*"

The methods were outlined reasonably well; however there were some points of confusion. It is possible that a cartoon schematic of the deployment of the DTS could be beneficial.

We show the layout of the fiber optic cable in Figures 3 and 4? Do you mean that you'd like to also see the location of the DTS and calibration baths? If so, we can add those to the figures.

The DTS itself had "two channels"and at times the wording became confusing because the DTS was deployed in a river channel.

We changed the wording to 'instrument channel' when discussing the DTS channels to avoid confusion.

Results largely consisted of summary statistics of the temperature data and modeled temperature. It would be instructive to do more analysis that relates to the Pacific salmon of interest and its 28 degrees C threshold. Additional investigation might be done to research the following questions: 1) What percentage of the deployment was 28 degrees C exceeded at the mainstem site? 2) When the temperature was exceeded, did the model over or underestimate the actual value and by what percentage? 3) Is there any way to go beyond the summary statistics of the dataset?

The conclusions reached are clear; however, they seem fairly straightforward in that a one-dimensional model would certainly miss fine scale variation that occurs in a real river channel. I think it could be shown more clearly and in further detail the times when the one-dimensional model succeeds or fails for the Pacific salmon.

Thank you for your specific suggestions. See the proposed 'Additional Analyses of DTS and TIR Temperature Variability' section above for our response.

How certain is that the results presented would not be different for a deployment time later in the summer? How might the results and discussion change for a period later in the summer?

We added the following text to the limitations section to address this comment: "*We deployed the DTS during mid-summer when we anticipated stream temperatures would be warmest (and when the DTS*

*was available) as worst-case scenario for thermal refugia and connectivity. Additional research is needed to quantify how results would change for deployments earlier or later in summer."*

Key numerical results could be added to the abstract.
We added numerical results of model fit with observed data and can add additional numerical results following the addition of new analyses suggested by all reviewers.

At times the writing does not appear to have a singular voice.
We read through the manuscript to improve writing and maintain a singular voice throughout the paper.

Specific Questions:
pg. 3, Line 30: In what way do the diversions and return flows influence the temperature? It could be beneficial to explicitly state they warm the waters.
Diversions and return flows usually warm the river, but do not always warm the river. For example, one of our findings is that water in the Wabuska Drain return flow can be cooler than the rest of the river. For that reason, we left our original sentence unchanged.

pg. 4, Line 10: What percentage of the deployment period did temperatures exceed 28 degrees C? Is this a common problem in the watershed (are there locations where it is always greater than 28 degrees C) or does the problem occur only during discrete heatwave events?
See proposed Additional Analyses of DTS and TIR Temperature Variability section above for our response.

pg. 4, Line 30: Is it necessary to describe how DTS works in this level of detail?
We omitted a couple sentences in this paragraph to remove unneeded details and improve readability.

pg. 8, Line 28: How were key features added to the model output?
We clarified that key features were georeferenced so that the modeled reached that contained the features could be identified. This section now reads:
*River features like agricultural return flows, diversions, beaver dams, and seeps were georeferenced so that the modeled reach that contained those features could be identified. Measured DTS and TIR temperature ranges were applied to the model outputs to estimate small-spatial scale variability within each 300 m modeled reach.* (starting page 9, line 30).

pg. 9, Line 29: It would be useful to show that there was not a large warming even that occurred between the two deployments at the two sites. Weather conditions should be ruled out as a factor of influence on the two deployments.
We will add a supplemental figure of air temperature data to show that there was not a large warming event that occurred between the 2 deployment events and to rule out weather conditions as an influence on the deployments.

Specific Notes:
pg. 2, Line 3: Therefore, assessing stream temperatures...
Text revised to this suggested wording.

pg. 2, Line 9: Stream temperature models are a useful tool for river management because they help...
Text revised to this suggested wording.

pg. 2, Line 23: ...spatially-continuous stream surface temperatures. However, it... What does "it" refer to?

Text revised to this suggested wording. We substituted 'TIR data' in place or 'it'.

pg. 3, Line 31: Interactions between... "between" compares two things while "among" compares more than two things

Thank you. We substituted the word 'among' in place of 'between'.

pg. 4, Line 12: Low instream flows from surface water diversions have also caused the Walker Lake level to decline

We kept our wording for consistency throughout of manuscript. We do not refer to Walker Lake as 'the Walker Lake' elsewhere in the paper.

pg. 5, Line 25: Verb tense was changed to present tense.

We double checked that our methods are described in past tense. Accuracy of data loggers are in present tense because it always holds true.

**Authors Response to Reviewer 2**

In this manuscript, high spatial resolution temperature observations are used to investigate small scale spatial variability in stream temperatures. These are placed on top of simulated stream temperatures with grid cells of 300m, leading to the conclusion that at moments when the simulated temperature exceeds the critical temperature (for LTC) of 28 C, there are always pockets of water that are cool enough to serve as refuges. Although these findings are potential useful, this manuscripts reads a bit as an engineering report with the focus on 'improving' an existing 1D 300m-resolution temperature model. There is little in-depth science involved: the results section is mainly a summing up of observations at the 300 m scale, while more quantitative assessment of the uncertainties/sensitivities of the used approach is missing.

Because I still think the available data is sufficient to improve the manuscript significantly, I am advising major revisions. What in my opinion is needed is a more in-depth analysis about spatial variability over a whole range of scales. For example: one can compare the left and right bank DTS measurements for each cross section as the smallest scale (_1m). Subsequently, the temperature range for progressively larger scales can be obtained. Finally, the implication for the model scale (300m) can be used as a practical case study. Similar things can be done for the TIR observations. Although it was not clear to me if the 10 points which were used to obtain the summary points were measured in 1 cross section or if they were measured over 300m, with a spatial resolution of 0.6 m, there seems to be enough data to do the same analysis for the TIR data.

Thank you for your detailed and specific suggestions. See the proposed 'Additional Analyses of DTS and TIR Temperature Variability' section above for our response and outline to improve our paper following your recommendations.

Line by line comments:
P2,L29: Westhoff et al. (2007) did their research in the Maisbich river in Luxembourg.

We apologize for our error. We have checked our references and revised this sentence to read: "DTS data were used to calibrate and validate a 1.3 km physically-based, one-dimensional stream temperature model of the Boiron de Morges River in southwest Switzerland (Roth et al., 2010) and a 580 m river reach in Luxembourg's Maisbich River (Westhoff et al. 2007)."

P3, L3: Neilson et al. (2010) did indeed not use these measurements, but they did model the temperature of transient storage zones.
This comment may be an error. We do not cite Neilson et al. 2010 on page 3, L3 and the Neilson et al. 2010 paper we cite is a different one from the reference the reviewer provided. We cite:

Neilson, B. T., Hatch, C. E., Ban, H. and Tyler, S. W.: Solar radiative heating of fiber - optic cables used to monitor temperatures in water, Water Resour. Res., 46(July 2009), 1–17, doi:10.1029/2009WR008354, 2010.

P3,L10: isn't spatial variability not the same as temperature range?
We included this phrase to specify that we quantify spatial variability as temperature range rather than other metrics like distributions. We changed the wording to "… evaluate small-scale stream temperature variability, quantified as the range of stream temperatures…," to clarify our intent.

P3, L23: please also indicate stream width and depth estimates.
We added width and depth so this sentence now reads "*The Walker River is a desert stream with mean annual flows of 15.5 – 30 m3/s, width of approximately 7.6 m and depth of about 32.9 cm.*"

P4, L31: change to: the majority of the reflected energy has it original wavelength.
Text revised to incorporate reviewer's suggestion.

P6, L25: Are these 10 locations spread over 300 m, or 10 locations covering one crosssection? And why are only values in the centre of the channel taken? Pools etc, are generally located on the sides of the channel. The spatial variability will probably be larger when these values on the side of the channel will be taken into account as well.
We will analyze the raw data versus using Watershed Sciences' summary points to address this comment. See further details in the 'Additional Analyses of DTS and TIR Temperature Variability' section above.

Section 3.4: Here, many statistical parameters are explained. As a reader I found it difficult to remember what the meaning is of each statistic. Furthermore, the symbols defined here hardly ever comes back in the results section. My suggestion here is to only define a few of them, and in the results section clearly state if a spatial or temporal range is discussed.
We followed your suggestion to remove many parameters. We clarified the statistical parameters that we kept. Overall this should clarify this section and remove some redundancy and unneeded detail.

P7, L16: use subscripts for TSavg: now it reads as T*S*a*v*g (the same for all other equations). Instead of 'avg' one can also use a bar on top of T
We have removed avg in the notation and use bars over the top of the variable instead.

P8, L1: what does "top of hour" mean?
We changed text to say 'hourly' instead of 'top of the hour' throughout the manuscript.

P8, L10: If I understood it correctly, these summary points were the mean, min and max. So why are there "on average" 3 summary points? This implies that at some location you have less and at others you have more summary points.

We will analyze the raw data versus using Watershed Sciences' summary points to address this comment.  See further details in the 'Additional Analyses of DTS and TIR Temperature Variability' section above.

P8, L15: L is river length, isn't it?
Yes.  We clarified the text to read "*where $\overline{T}_L$ is average stream temperature for the length of the East, West, or mainstem Walker River, L*".

P8, L28: What do you mean with "extrapolate model outputs"?
We re-wrote that section to clarify our meaning.  It now reads "*River features like agricultural return flows, diversions, beaver dams, and seeps were georeferenced so that the modeled reach that contained those features could be identified.  Measured DTS and TIR temperature ranges were applied to the model outputs to estimate small-spatial scale variability within each 300 m modeled reach*."

P8, L31: the key features are the same as the river features discussed in the previous lines, aren't they?
We changed 'key features' to 'river features' to clarify our meaning.

P11, L15: I guess you mean that the backwater location is subject to a higher thermal mass (or lower heat capacity)?
We revised this sentence to read "*The warmer temperatures occurred in an unshaded, shallow, backwater location subject to lower heat capacity*".

P11, L33: with a single snapshot in time one cannot conclude that the stream is cooling. One can only see that it is cooler downstream, but this can also be caused by warm water that only travels slowly downstream.
We revised this text to read "*Although Wabuska Drain was receiving agricultural returns and therefore contributing warm water, rather than the cool water observed during times with limited agricultural returns, a 4.5 km stretch of river downstream from the Wabuska Drain was 1 °C cooler than the segment of river upstream of Wabuska Drain*".

Section 5: Discussion: I miss a discussion on the difference between TIR and DTS measured ranges: which one has the largest range and highest temperature, and why? Although both methods measured at different periods in time, it must be possible to say something about it.
We added the following text to address this concern "*Overall, DTS measured a larger temperature range than TIR imagery in both the East Walker River and mainstem river (Tables 2 and 3) because the fine DTS spatial resolution could measure temperatures that varied spatially over short distances where beaver dams or return flows existed.  The warmest temperatures were measured by TIR imagery in the East Walker River, but by DTS in the mainstem, indicating that both methods are useful, and more widespread data collection of longer DTS deployments or repeated TIR collection would further improve results.*" (starting page 16, line 9)

P14, L4-5: Why is the change in temperature over time minimized on warm days? I would presume that on such days the rate of change is at its maximum, since also the daily amplitude is the largest on these days
We omitted to words 'warm, clear' to clarify that we minimized stream temperature changes through time by collected TIR imagery on days with similar weather conditions.

P14, L8-9: Can you quantify the effect of stratification?

We noted that quantifying temperature range from vertical stratification is outside the scope of this paper.

P15, L6: The definition of temperature range is defined as Tmax-Tmin, so it can never be a negative value.
We changed '*temperature range*' to '*temperatures*' to distinguish meaning from our earlier definition.

Section 5.4: this belongs in the conclusion (or summary) section.
We moved this section to the summary section.

Figures, general comment: add the caption of the figures also right below the figures: I know this will happen once the manuscript is published, but as a reviewer, I found it annoying that these captions were only listed in the text.
We will add captions to the figures when we submit a revised manuscript.

Fig. 3b: reduce the scale on the vertical axis to max 2 C?
We kept the scale on the y-axis to max 8 C so the figure is comparable with Figure 4b. We felt that different scales between Figures 3b and 4b would be misleading to readers.

Fig.5: In my opinion, there is little added value to the panels b-d. I also wouldn't call them "summaries" anyway: it is just a different way of plotting the same results.
We removed panels b-d from this figure.

Fig.7: What was the time for the simulated temperature? I am also wondering how variable the range over time is. This is especially interesting since the TIR data is from a completely different year than the DTS measurements. The legend of panel b can also be improved: What is the dark blue line, how are the average temperatures obtained?
We added the time for the simulated temperature to the caption and reworded it so it now reads "*Locations of river features that affect stream temperatures in the Walker Basin (a). Warmest predicted RMS stream temperatures for June 29, 2015 (6:00 pm) with estimated temperature ranges by river feature using DTS data from June 29, 2015 at the warmest observed time (3:15 pm) and TIR data from July 18 and 24 - 26, 2012) (b).*" We removed the average temperatures to simplify the figure.

The DTS measurements give spatio-temporal ranges and the TIR measurements give spatial variability for one time period. TIR and DTS data were acquired at different times. We have overlain both DTS and TIR onto the same model run because it reduced data/complexity and led to largely the same results. We could overlay these data onto the RMS model output for matching time periods (i.e., TIR for 2012 model run and DTS for 2015 model run) and separate Figure 7b into 2 figures if this is misleading. We like this figure because it is our perception that managers make decisions as new data is analyzed and shared without always developing contracts for additional model runs. But as we've said, we can omit it if reviewers feel it is misleading.

**Authors Response to Reviewer 3**

Hydrology and Earth System Sciences manuscript HESS-2018-441 describes the result of a study to quantify small-scale temperature variability in the Walker River, Nevada, using a combined approach leveraging DTS, thermal infrared (TIR) remote sensing and one-dimensional river temperature

modelling. DTS and TIR data were incorporated with coarser temperature model predictions to identify potential thermal refugia habitat that may otherwise be missed from lower-resolution temperature model outputs. While the general idea of combining TIR, DTS and model predictions is laudable (and there is definitely a need for this type of research), I felt that the manuscript was generally too descriptive, with a lack of formal hypothesis testing or deeper analysis. Furthermore, the method by which the TIR/DTS data was combined with the temperature model predictions (through simple addition of temperature ranges to the temperature model-derived thermal long profiles) was rather simplistic and does not appear to offer a substantial benefit over the use of the DTS/TIR data alone. Despite these issues, I support the general concept of the manuscript and believe that the idea of combining DTS, TIR and model predictions has merit, but I feel that significant revisions (including a much more in-depth treatment/analysis of the data beyond the scope of what would normally be considered 'major revisions') is needed prior to publication.

*Please see the proposed 'Additional Analyses of DTS and TIR Temperature Variability' section above for our outline to improve data analysis in our paper.*

—General comments—
-In section 3.4 (and table 1), it is quite difficult to remember/follow the names of the different variables. Is there any way of streamlining/simplifying this?
*We removed many parameters and clarified the statistical parameters that we kept.  Overall this should clarify this section and remove some redundancy and unneeded detail.*

-It's probably not necessary to give all of the detail about the DTS system (ie. 2 channels, cable stress, etc) in the methods and results sections. Consider removing all superfluous information to streamline the manuscript.
*We removed unneeded DTS details and text to streamline the manuscript.*

-It would be nice to see some of the TIR imagery to allow for comparison with the DTS data; given the large time difference between the TIR and DTS data, the information in table 3 is of limited use.
*We kept Table 3 since it provides a succinct summary of max, min, and average TIR measured temperatures and ranges.  We will add TIR imagery to the revised manuscript (see the additional analyses section above).*

-Use of the word 'RMS' (ie. the name of the model) is confusing in places when you are also talking about RMSE. Consider putting 'RMS' in italics or similar to avoid confusion.
*We italicized 'RMS' throughout the manuscript to avoid confusion with RMSE.*

-There is very little information about the RMS model in terms of inputs, implementation, etc. I appreciate that this information is already given in the other references provided, but it would nonetheless be useful for see it outlined in this MS (maybe a short paragraph explaining model function, input data, etc).
*We improved the description of the RMS model to include description of inputs, model function, and implementation.  Section 3.3 now reads:*

> *"RMS is a 1-dimensional hydrodynamic and water quality model which solves the St. Venant equations for conservation of mass and momentum and the Holly-Priessmann mass transport equation (Hauser and Schohl, 2002).  RMS has a hydrodynamic module and a water quality module, which are run sequentially (Hauser and Schohl, 2002).  Input requirements for the hydrodynamics module are channel geometry, roughness coefficients, boundary conditions and initial surface water elevations. Outputs are velocity and depth at each model node which are input into the water quality module.  Additional inputs*

*for the water quality module include weather data, riparian shading estimates, boundary temperatures and initial water temperature. Outputs are hourly stream temperatures.*

*Previous research provided modeled streamflows and stream temperatures for one wet (2011) and three dry (2012, 2014, 2015) irrigation seasons (April 1-October 31) (Elmore et al. 2016; Null et al. 2017). The model was developed to simulate stream temperatures from environmental water purchases that alter thermal mass to improve habitat for native organisms and connect Walker River and Lake. Irrigation season (April 1–October 31) was modeled because that is the time period that environmental water purchases occur from irrigators. A total of 305 km of the East Walker, West Walker, and mainstem Walker Rivers were represented in RMS at an hourly time step and 300 m modeled reach spatial resolution. As a 1-dimensional model, each reach has a homogenous temperature throughout the reach. Walker River modeled extent includes the East Walker River downstream of Bridgeport Reservoir (river km 243 to 117), the West Walker River downstream of Topaz Reservoir (river km 60 to 0) and the mainstem Walker River to Walker Lake (river km 117 to 0) (Fig. 1). For additional model detail see Elmore et al. (2016) and Null et al. (2017)."*

-It's not immediately clear in the manuscript how you 'apply' the DTS/TIR temperature data to the model. From my reading, you simply add/subtract the temperature range to the model outputs. Is this correct? If so, I would have thought that this process (ie. calculating temperature variability from DTS and TIR and simply adding/subtracting to/from the model outputs) is accompanied by a range of issues given that the DTS data is essentially a measure of spatio-temporal temperature range, whereas the TIR data only gives spatial variability. Also, given that the TIR data and DTS data were acquired at different times, it does not seem appropriate to overlay these data onto the RMS model output for the same point in time (eg. fig 7).

We changed this wording to "*Measured DTS and TIR temperature ranges were added to model outputs to estimate small-spatial scale variability within each 300 m modeled reach. This provided an estimate of spatial variability missing in the model output that is needed to gain insight into potential habitat availability at smaller-spatial scales.*" The DTS measurements give spatio-temporal ranges and the TIR measurements give spatial variability for one time period. TIR and DTS data were acquired at different times. We have overlain both DTS and TIR onto the same model run because it reduced data/complexity and led to largely the same results. We could overlay these data onto the RMS model output for matching time periods (i.e., TIR for 2012 model run and DTS for 2015 model run) and separate Figure 7b into 2 figures if this is misleading.

-What is the reason for the large temperature discrepancy between the two river banks? This kind of information is quite interesting and could benefit from a thorough treatment in the manuscript.

Thank you for this suggestion. We will add it to the revised manuscript. See the 'Additional Analyses' section above.

-Section 3.4 of the methodology is difficult to follow. Given that this section contains the majority of the analyses covered in the results and discussion sections, it would be beneficial to restructure this section to make the subsequent results and discussion easier to follow.

We completed a major revision of this section, removing redundancy and unnecessary detail. We hope this section is easier to follow now.

—Introduction—
P2 L1: Would be good to have 1-2 sentences of more general information on the importance of river temperature in the context of climate change.

We added the following phrase to the 1st sentence: "…and climate change is anticipated to increase stream temperatures in summer, fall, and winter (Isaak et al. 2012)."

P2 L24: It would be a good idea here to qualify the point about TIR only providing a single snapshot in time by saying something along the lines of 'unless acquired multiple occasions'.
We added this phrase.

—Methods—
P4 L22-28: These lines are potentially redundant and could be moved or redistributed elsewhere in the methods section.
We omitted these lines and added the pertinent information that was not redundant elsewhere in the methods.

P4 L30-31 and P5 L1-5: I'm not sure that a description of how DTS works is necessary; consider removing this paragraph (or at least, shortening it to one sentence)
We shortened this section so it is now 2 sentences: "*DTS units measure temperatures by sending a laser pulse down a fiber-optic cable and timing the return signal. Although the majority of the reflected energy has its original wavelength, a portion of the energy is absorbed and re-emitted at both shorter (Anti-Stokes backscatter) and longer (Stokes backscatter) wavelengths. Temperatures along the cable are determined from the Stokes/Anti-Stokes ratio (Selker et al. 2006).*"

P5 L6-13: Consider adding a figure to illustrate the deployment of the DTS. I appreciate that this is partially covered in fig 1 (large scale) and fig 3/4, but it would be nice to see a full resolution map showing the DTS installation.
We show the layout of the fiber optic cable in Figures 3 and 4? Do you mean that you'd like to also see the location of the DTS and calibration baths? If so, we can add those to the figures.

P5 L18-22: Is this information necessary? Consider removing.
We omitted lines 19-22 (line numbering from original document).

P5 L28: This sentence formulation is slightly difficult to follow. Do you mean to say that the calibration process consisted of using a linear transform to correct the DTS based on the difference between the DTS and thermocouple temperatures in the ice bath?
We revised this sentence to read: "Calibration used a linear transformation to correct the DTS data based on the difference between the DTS and thermocouple temperatures."

P6 L18-32: There isn't any mention here of the winter TIR data collection flights which you subsequently refer to later on in the manuscript. The manuscript only appears to have dates and hydrometeorology information for the summer flights. I appreciate that the winter data is only used for locating seeps, but it would be good to talk about the winter flights here first.
We discuss the winter and summer flights in the first sentence of Section 3.2.

P6 L32: Although I managed to find the Watershed Sciences documents online, they were quite difficult to track down. It would therefore be good to give some further brief details of the flights, for example RMSE or R2 of TIR data vs. logger values, etc.
We added the following sentences to describe TIR validation. "*TIR radiant temperatures were validated with 28 Hobo Pro and iButton kinetic sensors. For the river extent used here, TIR data were within 0.5 °C of the instream sensors except for one location in the East Walker River where two instream sensors*

*were 1.7 °C and 3.3 °C cooler than radiant TIR temperature, and one location in the West Walker River where an instream sensor was 1.1 °C cooler than radiant temperature."*

P7 L7: Sentence ('each reach [: : :] throughout the reach') is a little clunky – consider rewriting.
We revised this sentence to read: "As a 1-dimensional model, each reach has a homogenous temperature."

P8 L28: This sentence ('To extrapolate model outputs: : :') is difficult to follow; it is difficult to understand exactly what you are doing here. Can you think of a different way to explain this step in the methodology?
We've revised this sentence to read "River features like agricultural return flows, diversions, beaver dams, and seeps were georeferenced so that the modeled reach that contained those features could be identified."

P8 L32 and P9 L1: What do you mean by 'were developed in 2012'? Do you mean that diversions/return flows were implemented in 2012 in the model, or in reality? Or just that they were mapped/identified?
We changed wording from 'developed in 2012' to 'identified in 2012'.

P8 L2-4: If you wanted to exhaustively identify all seeps, I wonder if a better practice would have been to combine data from the winter and summer survey flights? Also, please give some detail about how the seeps were identified. Was it from manual photo interpretation? Were aerial photos acquired simultaneously to aid the interpretation process?
Due to the low water in July, the majority of locations on the Walker that showed groundwater activity in the winter were dry at the time of the summer flight. Seeps were identified from cooler stream temperatures that could not be attributed to shadows, cutbanks, or vegetation. We added text describing this to Section 3.4.

—Results—
P9 L26-30: These sentences would be better suited to the discussion section.
We moved the last sentence to the Discussion (section 5.2). We left the sentences about the dam releases so the results make sense to readers unfamiliar with this system.

P10 L20: I'm not sure what you mean here when you say that the daily max/minimum temperature changed little when analysed with the Walker River. Do you mean to say that a lack of large-scale temperature variability in the study reach masks considerable localised variability in areas like the Wabuska drain?
Yes. We revised the sentence to reflect your suggested wording. "When the 20 m section of the Wabuska Drain return flow canal (shown approximately at distance 110 – 175 m in Fig. 2b) was analyzed with the mainstem Walker River, daily minimum and maximum temperatures did not change because reach scale temperature variability masks localized variability in areas like the Wabuska Drain."

P10 L25-30: Some of this material might be better suited to the discussion section.
We moved this text to Section 5.2 in the Discussion.

P11 L32-33 and P12 L1-5: This would also be more suitable to the discussion.
We moved this text to the discussion.

—Discussion—

P13 L23-31 and P14 L1-9: I'm not really sure what information this 'limitations' section adds to the manuscript. There are clearly limitations when conducting an approach such as this combining TIR, DTS and modelling. However, this section reads more like a list of problems associated with TIR and DTS data collection (which have already been well established in the literature) rather than a critical appraisal of the inherent difficulties of combining and comparing these types of data with 1D temperature models, which would be much more interesting (and potentially useful for the reader).

We largely rewrote the limitations section.  We briefly summarized problems associated with TIR and DTS data collection which have been previously discussed in the literature.  Then we highlighted the difficulties of comparing datasets with different spatial and temporal resolutions and that were collected in different years.

We added the following text: "*Obtaining small-scale spatial and temporal stream temperatures and comparing them to model results has a number of limitations.  First, the spatial and temporal resolution vary between DTS and TIR datasets and with simulated model results.  TIR imagery represents a single point in time unless costly flights are repeated.  DTS measurements are dense (1 meter in these deployments) with a 15 minute temporal resolution, but are limited by cable length and field crews to monitor the deployment.  Comparing measured data to hourly model results with 300 m spatial resolution further reduces the number of comparable observations.  Second, DTS and TIR measurements were collected in different years because we used existing TIR imagery collected as part of the Walker Basin Project, a multi-partner comprehensive effort to sustain the basin's economy, ecosystem, and lake. Future studies could collect data specifically to overlap in time and space; however, opportunistically using existing data for re-analysis and to improve model result interpretation and river management is a laudable goal that may reduce the cost of river science and management.  Multi-year, multi-partner river monitoring, modelling, and management is common in large, important, or complex river basins.  This research highlights the differences in temperature variability given alternative sampling methods.*"

—Figures—

Figure 7(b). As discussed above, it is not clear how you combine the RMS stream temperature data with DTS data from June 2015 and TIR data from July 2012. Surely this mixing up of different dates and times means that the temperature ranges from the DTS cannot be comparable to those from the TIR? Also, to what does the 'average temperature' refer? Is it from the DTS data (temporal) or is it average temperature (spatial) calculated as the mean of all pixels covering the refugia/beaver dam, etc?

The DTS measurements give spatio-temporal ranges and the TIR measurements give spatial variability for one time period.  TIR and DTS data were acquired at different times.  We have overlain both DTS and TIR onto the same model run because it reduced data/complexity and led to largely the same results. We could overlay these data onto the RMS model output for matching time periods (i.e., TIR for 2012 model run and DTS for 2015 model run) and separate Figure 7b into 2 figures if this is misleading.  We removed the average temperature to clarify the figure.

**Authors Response to Reviewer 4**

The article compares temperature data from DTS and thermal imagery within a stream reach where one-dimensional temperature modelling was conducted. The study takes place in the Walker Basin, NV, where stream temperatures can exceed thermal tolerance for native, threatened trout. This study provides a unique combination of methods and is useful in understanding the pros and cons of each. The study is relevant for work that seeks to restore habitat for fish in streams where temperatures rise above the thermal tolerance of threatened cold-water species.

There are a few places where the manuscript could be improved. The authors generally summarize the differences between the methods, but they could take it to the next level with doing a more quantitative analysis comparing the different methods. At the end, there is not a strong conclusion, or list of pros and cons, comparing DTS vs TIR methods for validating or contributing to stream modelling efforts. What would the authors decide to use - DTS or TIR - based on the results of this paper?

Second this paper has the data to conduct analyses that look at the availability of different water temperatures that are relevant for trout. For example, the authors focus on 28C as a thermal cut off for LCT, but these fish may become thermally stressed at much lower temperatures (e.g., 21 or 22C). Additionally, very cold temperatures in the summer may not be ideal for growth. The authors could consider adding an analysis that describes the area of stream that is available to LCT below different cut offs, e.g., below 18C, below 21C, below 25C, and below 28C, and the distance between those areas. The DTS and TIR methods identify river features that the one dimensional model doesn't, but how important/large/cold is the water provided by these features? The authors could also compare how the different methods perform in quantifying the amount and connectivity of thermal refugia.
Thank you for your detailed and specific suggestions. See the proposed 'Additional Analyses of DTS and TIR Temperature Variability' section above for our response and outline to improve our paper following your recommendations.

One way that could improve the flow of the results section would be to put the comparison of all three methods up front, and then describe the results of how temperatures differ in the basin second. These are currently split into different sections, with the temperatures differences in the basin described following each method.
Thank you for this suggestion. We will re-evaluate and improve the organization of our paper following completion of the additional analyses of DTS and TIR data.

Additionally, the description of the DTS channels can be a bit confusing because there are river channels and DTS channels – is there another descriptor that could be used in place? The focus on the Wabuska diversion can also be confusing, because it was flowing during the time window of one method and not flowing for the time window of the other method.
We changed the wording to 'instrument channel' when discussing the DTS channels to avoid confusion. The Wabuska Drain is important to our findings, so we left it in the manuscript. We have tried to be clear about when it was flowing and when it was not flowing, but had standing water.

Finally, the introduction and discussion could use more context for why micro thermal regufia is important for cold water fish and trout. The author should look for more recent citations of Sutton et al. 2007, Brewitt and Danner 2014, etc.
We added the following text and citations to the introduction: "*Trout and salmon avoid heat stress by sheltering in pockets of cold water when stream temperatures are near upper thermal tolerances (Dunham et al. 2003; Sutton et al. 2007) and climate change is anticipated to increase stream temperatures in summer, fall, and winter (Isaak et al. 2012). Recent research has quantified when and where cold-water fish need thermal refugia (Brewitt and Danner 2014), estimated the size of thermal refugia and the distance between refugia (Fullerton et al. 2018), demonstrated how fish use thermal refugia (Frechette et al. 2018), and measured the length of time that fish can survive between refugia (Pepino et al. 2015). Where stream temperatures are warming or where cold-water fish species are at the southern extent of their range, assessing stream temperatures at small temporal and spatial scales is*

*thus important to quantify stream temperature heterogeneity and best manage complex and variable riverine habitats (Vatland et al. 2015)."*

We also added the following citations to the discussion: "*Future research is needed to validate temperature ranges by river feature at the watershed-scale, evaluate how fish use thermal refugia, and to improve understanding of the resiliency of thermal refugia with anticipated climate change (Fullerton et al. 2018; Frechette et al. 2018; Stevens and DuPont 2011; McCullough et al. 2009)."*

Specific Comments:
Pg 2 Line 5. It could be useful to explain here explicitly what you mean by one dimension. (Obvious to stream temp. modelers but less intuitive to managers). Do you mean longitudinal interpolations along the stream?
We changed this sentence to "However, one-dimensional stream temperature models *that estimate longitudinal stream temperature changes and* that are applied at the watershed-scale are poor predictors of thermal micro-habitats" to describe one-dimensional stream models for readers unfamiliar with modeling.

Pg 2 Line 13-15. Similar comment as above – what does 2, vs 3 dimensions mean for streams? Length, width, depth, and in that order?
1-, 2-, and 3-dimensional stream temperature models typically represent longitudinal, lateral, and depth dimensions, in that order.  To clarify, we revised the sentence to "… one-dimensional because they are less data intensive and more computationally efficient than two- or three-dimensional models *that also vary by width and depth*…".

Pg 2 Lines 23-25. Switch the order of these sentences. Describe how TIR has been used to locate various stream features, but then the downside is that it is a single snapshot in time.
Done.

Pg 3 Lines 2-5: This sentence is key in setting up what your study contributes to the ones that you cite, but it isn't clear. These methods have been used to calibrate reach scale models but hasn't been used to quantify temperature ranges within model reaches? The difference in the wording is very slight, and needs to be clarified.
Yes, we mean that DTS and TIR have been used to calibrate models, but haven't been used to quantify temperature ranges within the model reaches.  We have rephrased the sentence to clarify our meaning: "*While DTS and TIR have been used to calibrate reach-scale models, no studies have used DTS and TIR to quantify temperature ranges by river feature within model reaches, and use to that information to estimate likely temperature ranges at the watershed scale.*"

Pg 3 Lines 9 – 10 Objective #3 is has circular wording, it's not clear what the objective is. Is the goal to identify features with greater temperature ranges because they have variable temperatures? Maybe the objective is just to identify those features?
The objective is simply to identify the features.  We omitted the phrase "…*due to spatially variable temperatures*" to clarify meaning.

Pg 3, Lines 11-18 I suggest reducing this description of theWalker Basin to 1 sentence, maybe 2 sentences here. You get into more detailed description in the next section. For example, environmental water purchases seem relatively uncommon, and you do them more justice in the next section.

We omitted 2 of the sentences so now we have 2 sentences here: *"Nevada's Walker Basin was the study watershed and is representative of other arid and semi-arid watersheds in western USA where cold water species, like trout and salmon, are temperature-limited. River restoration is ongoing in the Walker Basin and there is a clear need to understand small-scale stream temperature ranges in different different river features (e.g., pools, confluences) to identify temperature barriers to migration and better interpret watershed-scale model results."*

Pg 4 Second paragraph – It would be nice to include some description about how narrow the range of LCT is. It makes your study more special.
We added the historical range of LCT. This sentence now reads: *"The historical range of LCT is the Lahontan Basin in eastern California, southeastern Oregon, and northern Nevada*, although LCT are limited in their native range by warm stream temperatures, low streamflows, and low dissolved oxygen in the Walker River (Coffin and Cowan 1995; USFWS 2003)."

Pg 4 Line 8 – Are all these study sites in CA? For a European journal, the places should be better located.
These sites are in Nevada and Oregon. We've added the state in which they're located to the text.

Pg 4 Line 14 - This sentence is a little unclear as to what it means in context of the previous sentence. Because the lake is inhabitable, it means the lake and stream systems are disconnected?
We deleted this sentence.

Pg 4 Line 17 – Restore to tolerable salinity ranges? Restore is a broad word. The discussion of salinity is really interesting, but perhaps not relevant to your stream temperature focus.
We revised the sentence to "to restore Walker Lake to salinity to tolerable levels". We kept the discussion of salinity and Walker Lake to add context to the research.

Pg 4 Line 22 – Can you be more specific about what a 'dry year' means in this basin? I.e., received <25% of historical rainfall, or stream base flows were at XX% of the average flows for that time of year? (Especially since you do have USGS gauge data)
We removed this section following the recommendation of another reviewer. However, we added that in dry year 2012 snowpack was 50% of normal and in dry year 2015 snowpack was 5% of normal to the DTS and TIR subsections.

Pg 4 Line 26 – Comparing measured and modeled data? (rather than between)
We changed "between" to "comparing".

Pg 6 Line 31 – the East Walker has more flow than the mainstem – presumably because of diversions? Remind the reader of that again
Diversions vary streamflows in both the East and West Walker Rivers so that some days flows are higher in the East Walker River and some days they are higher in the West Walker River. We do not want to imply that West Walker flows are always lower than East Walker flows, so we made no changes to the text.

Pg 9 Line 30 – was the Elmore et al. 2015 study also conducted in the Walker R basin? This point may be better situated in the discussion.
We moved this sentence to the Discussion (section 5.2).

Pg 9 – Fig 2 caption – A reminder of what is the Wabuska Drain (not-flowing, but standing water from an ag ditch) would help the reader in the caption.

We added a description of Wabuska Drain, so the caption now reads: "Stream temperatures measured for the length of the DTS cable at East Walker River (a) and mainstem Walker River (b) DTS sites. Wabuska Drain, *which was not flowing but had standing water during sampling*, is located at cable distance 110-175 m in the mainstem Walker River site (b)."

Pg 10 – Fig 3 – The sub-panels in Fig 3 should be labelled. In looking at the figure, it's not clear why these two features are called out. It looks like there is more variation between river left and river right than anything else. Fig 3 b could probably be moved to supplemental.

We amended Figure 3 caption to include that insets show details of spatial temperature variability. When we complete the analysis of temperature differences between the right and left banks we can change the inset or remove the downstream-most inset, if needed.  We moved figure 3b to supplemental information.

Pg 10 – Why do Fig 3 and Fig 4 visualize different times, 5:30 pm vs 3:15 pm?

Figures 3 and 4 visualize each reach at their maximum daily temperature.  We felt this was more representative and comparable than picking an arbitrary time that is the same for both reaches.

Pg 10 Lines 19-31 – How different would this look if the Wabuska drain was flowing? Is it more often flowing or not flowing in the summer months?

We do not know how different results would be if the Wabuska Drain were flowing.  Thus we added a line to the limitations section: "*Additional research is needed to quantify how results would change when the Wabuska Drain is flowing, or for deployments earlier or later in summer.*"  The Wabuska Drain is usually flowing during summer, except during dry/very dry years when irrigators use more groundwater than surface diversions.  The majority of years in the past decade have been dry/very dry years.

Pg 10 – Fig 4 – It would be nice to add a line or circle indicating where the beaver dam is on that reach of stream. Fig 4 b could also be moved to supplemental. However it might be worth exploring why there is day-to-day variation in the temperature ranges – could plot the data against air temperature range for the day?

When we complete our additional analysis we will add them to the map to label the beaver dam locations.  We moved figure 4b to supplemental information.

Pg 11 Lines 10 – 11 Have LCT been observed using the Wabuska drain when it is flowing, or when stream temps aren't so high at the mouth?

LCT have not been observed near the Wabuska Drain that we are aware of.

Pg 11 Lines 31-32 Are there studies that show that LCT cannot move through warm temperatures? Maybe save the temperature implications for fish for the discussion, when you can do a more thorough review of studies on LCT (and related salmonids) and their thermal tolerance. They may be able to move through a small area of warm temperatures. What may be more important is the extent of cold water refuges, which could be quantified with the given data.

We moved all implications for fish to the discussion so that we could put the findings into context with the literature.

Pg 11 – Lines 33-34 – Make it clear that the cool water from Wabuska was observed with the other method/time window when stream temps were monitored.
We added 'during the TIR flight' to the sentence to clarify.

Pg 12 – Fig 5 – Similarly to the other figures, the sub-panels b,c,d could be moved to supplemental since they are redundant with the main figure. Alternatively, they may be a better way to present the data in the main figure, so you could consider dropping panel a.
We removed panels b-d from this figure.

Pg 13 – Lines 16-19 – Model estimated within 1C – that's pretty good! But under estimating by 2.5C is less desirable – could be a point to discuss.
We discuss this in section 5.3 where we add "*TIR stream temperature measurements in the lower reaches of the mainstem Walker River were considerably warmer than simulated results and remain near LCT lethal thermal threshold for an additional 45 km than was previously estimated*."

Pg 14 – Lines 13-15, Maybe it didn't exceed 28C, but trout can be thermally stressed at much lower temperatures.
We added that trout can be thermally stressed at lower temperatures to the sentence.

Pg 14 Line 23 – Is this study from the Walker R Basin, or generally citing groundwater depletion from drought?
This study is from the Walker Basin.  We made no change since the sentence specifically discusses interflow to the Wabuska Drain.

Pg 14 Line 30 – How migratory are LCT? Would they historically migrate between lakes and streams, and was that during the summer? If not, then you could shift your focus to movement and opportunities for longitudinal connectivity rather than migration.
LCT historically migrated between Walker River and Lake. However, we changed this sentence to focus on aquatic habitat longitudinal connectivity because it is more broadly applicable for cold water habitat and species.

Pg 15 Lines 9-10 Describing that there is a range in temperatures doesn't necessarily mean that there is thermal refugia. How cold is the water around these features? If it has high variability but it still warm, then it may not provide refuge for LCT.
We added and subtracted the temperature ranges that we measured from the modeled values.  It doesn't mean there is always thermal refugia, but is suggests there may sometimes be refugia.  We modified our wording to "*suggests that cool-water refugia may sometimes exist*" so that we do not oversell our findings.

Pg 15 Lines 27-29 This would be a good place to give a nod to the work that has evaluated how fish use thermal refugia (do a forward search on Sutton et al 2007).
We added 4 references (Fullerton et al. 2018; Frechette et al. 2018; Stevens and DuPont 2011; McCullough et al. 2009) to give a nod to current research on how fish use thermal refugia.

---

## Author Response (AR2)

**Response to Reviewers**

Thank you to the reviewers and editor for careful review of our paper. We are confident that it will improve the paper. Below we provide detailed responses to every comment. Reviewers' comments are in black and our responses are in blue.

**Authors Response to Reviewer 1**

While this paper has made improvements from the previous version, there is still room for further improvements. Most of my line-by-line comments here are on improving comprehension for the reader. One way the authors could do this would be to relate the initial four objectives that are laid out to the methods and results sections. Some clear topic sentences and hand-holding sentences would go a long way in demonstrating how the organization of the paper is related to those topics.

We combined objectives 1 & 3 per your suggestion below. Then we re-organized methods and results and included subheadings that relate to each objective to clarify our work for readers. In doing this, we moved the sections about estimating thermal habitat and refugia connectivity (objective 3) from the discussion to the results. We also gave the paper a detailed read-through, adding segues and topic sentences to improve readability. Finally, we updated the title of the manuscript to reflect that focus changed away from river restoration and toward understanding thermal refugia with DTS, TIR and modeled datasets through the peer-review process.

Similarly, the figure captions could use more detail on which data is being presented, since there are many datasets it is easy to get lost as the reader.

We double checked all figure captions and clarified data sources where needed.

The paper still reads as a report summarizing findings of the methods, without clear hypothesis testing.

We removed extra details regarding DTS and TIR findings, focusing instead on methods and results that tie to our research objectives. We also rewrote the discussion to tie our findings to the literature and discuss thermal refugia in the Walker River for Lahontan cutthroat trout (see next comment).

Lastly, the authors did make an effort to summarize the data for LCT temperature ranges, but I think these analyses could be expanded. Can the authors calculate the spatial extent of different temperatures? How about the connectivity of cold-water patches? How does this build on previous work that has studied thermal refugia – are the features of thermal heterogeneity unique to this system, or have they been observed elsewhere?

We added the connectivity of thermal refugia in this basin (pg 13, ln 19-21): "The shortest distance between refugia, or cooler pockets of water, was 0.3 km, which was the spatial resolution of model reaches. The maximum distance between refugia was 37 km and occurred near Weber Reservoir in the mainstem Walker River. The mean distance between refugia was 2.8 km and the median distance was 0.9 km." We also rewrote the discussion to focus on how our research builds on previous work and highlight how our method quantifies thermal refugia connectivity using modeled and high-resolution measured data.

Abstract
-Could you include some of the results of how temperatures relate to LCT thermal tolerances in the abstract? This might broaden your readership and citations from that audience.

We revised the abstract to highlight LCT thresholds and thermal refugia results to broaden readership.

Introduction
Pg 2 Lines 20-21– Citation for this? Some small scale models do, but perhaps not at a watershed scale?

We added 'watershed-scale models' to qualify the sentence, and cited Null et al. 2017, who discuss these limitations of watershed-scale one-dimensional modeling.

Pg 2 Line 25 – remove 'spatial and' (redundant with point locations)
Done.  Thank you.

Pg 3 Lines 13-14 – Can you use data to corroborate a calibration, or do you mean just corroborate the model (drop calibration)?
We removed the word calibration.

Pg 3 Lines 12-15 – As written, it is not clear to the reader at this point what the difference is between Objective 1 and Objective 3.
We combined objectives 1 & 3 into a single objective (#1).  Our objectives now read (pg 3, ln 20 – 24): "The objectives of this study were to 1) evaluate stream temperature variability, quantified as the range of stream temperatures, at multiple spatial scales and by river feature using DTS and TIR imagery, 2) use those data to corroborate an existing one-dimensional, 300 m spatial resolution, watershed-scale stream temperature model, and 3) add measured, spatially explicit stream temperature ranges to model results by river feature to estimate thermal habitat and thermal refugia connectivity throughout a watershed."

Pg 3 Line 16 – further (type-o)
Fixed.  Thank you.

Pg 3 Line 20 – You may mean barriers to movement or connectivity, rather than migration here ("migration" has very specific implications)
We changed 'migration' to 'movement'.

Pg4 Lines 2-3 –The "Walker River" in the first part of the sentences refers to which part of the river? (At what point in the watershed is the citation referring to/how does it differ from the "mainstem Walker River" in the second part of the sentence?
We omitted this part of the sentence as it is tangential.  In general the mainstem Walker River is a losing system.

Pg 4 Lines 11-12 – This sentence might be improved with a little more context – e.g., measure stream temperatures exceeded the 28C threshold (frequently? In many places?) in the 2014-2015 summers, demonstrating that warming stream temperatures are a concern for LCT in the Walker basin.
We rewrote this sentence to read: "Measured mainstem Walker River stream temperatures exceeded the acute 28 °C temperature threshold for LCT throughout summer in 2014 and 2015, demonstrating that warming stream temperatures are a concern for LCT in the Walker Basin". (pg 4, ln 17 – 19)

Pg 4 Line 20 – Improve habitat conditions for who?
We added 'for Lahontan cutthroat trout and other aquatic biota' to this sentence.

Pg 5 Lines 7-9 – If none of your data was from these double-ended set of measurements, do you need to include this in methods?
We removed these sentences.

Fig S2 – Discharge on the Walker doubles from June 26 – June 28 – what was the reason for this change in stream flow?
There was no measurable precipitation during the DTS deployment.  The change in streamflow was from upstream reservoir releases, which is described in the supplemental information.

Pg 6 Lines 12-14 – It's not clear what the summary points are, exactly, here. Which surface inflows were

considered, all of the tributary confluences and ditch return points?
We clarified this and it now reads "Watershed Sciences, Inc. also provided summary point data, which are minimum, median, and maximum temperatures of 10 pixels from the middle of the stream." (pg 7, ln 15 – 16)

Pg 7 Line 5-6 The restoration goal of water purchases is well described above, you could exclude it here
We removed this sentence.

Pg 7 Line 12 – details (plural)
Corrected.

Table 2 caption – It's not clear from the caption why data is presented from just afternoon/morning of different days, and what deployment/demobilization days are
We want to highlight that on some days data was not collected for the full day. We changed the caption to read "Daily stream temperatures and ranges for DTS deployments in the East Walker River (11:15 on 6/19/15 to 9:45 on 6/23/15) and mainstem Walker River (14:15 on 6/25/15 to 12:30 on 6/30/19)."

Pg 8 Line 8 – How did the flight times determine which data to use?
We removed that sentence as we think it is clear that modeled and TIR data at corresponding times and locations were compared.

Pg 9 Line 2 – Was percentage evaluated by time or space?
What about the spatial connectivity of suitable temperatures along a stream?
We changed wording to clarify: "The percentage of time that DTS and modeled stream temperatures were below 21 °C, 24 °C, and 28 °C, and the river extent that TIR and modeled stream temperatures were below the same thresholds were also calculated."

In the 2nd paragraph of section 4.4, we added results for the spatial connectivity of suitable temperatures along a stream: "Adding observed DTS and TIR temperature ranges from modeled results indicates that cool-water refugia may sometimes exist to support species migration between Walker Lake and tributaries of the Walker River (Fig 9b). The shortest distance between refugia, or cooler pockets of water, was 0.3 km, which was the spatial resolution of model reaches. The maximum distance between refugia was 37 km and occurred near Weber Reservoir in the mainstem Walker River. The mean distance between refugia was 2.8 km and the median distance was 0.9 km." (pg 13, ln 17-21)

Pg 12 Line 11 – Couldn't it be possible to edit out the pixels that contain riparian areas? Clip the buffers to the stream water extent? This would make your comparison of TIR closer to the other stream temp methods
There is substantial uncertainty as to where the water extent is. In other words, it is sometimes unclear whether pixels represent vegetation, shallow water, bare soil, or combinations of all three surfaces. We do not have visible imagery that corresponds to the same time period as the TIR imagery. It is thus a time-consuming exercise that will produce uncertain results. For that reason, we chose not to edit out riparian areas and focused on minimum temperatures instead.

Fig 8 – I find this stacked bar chart to be hard to interpret, personal preference.
We omitted Figure 8. The same information is presented in Table 5.

Pg 13 – Consider adding a section at the beginning of the discussion summarizing your key findings and interpretations, before diving right into limitations.

We moved the summary of our key findings to the first paragraph of the discussion to follow standard paper organization.

**Authors Response to Reviewer 2**

The authors have improved their manuscript according to all reviews, although I sometimes still have difficulties to distinguish the different statistical measures. Besides that, a couple of new issues have been raised as well, while every now and then I would like to see some more explanation about what exactly is done.

The main 'new' issue comes from the newly mentioned literature listed in the introduction on P2, L4-7. Although I am not familiar with this literature, it is stated here that already a lot is known about how, where and when refugia are needed. This means that this data could be applied to the results presented in this manuscript, but this is unfortunately not done. Instead, the authors state that (p16,L10-13) "Future research is needed to validate temperature ranges by river feature at the watershed-scale, evaluate how fish use thermal refugia, and improve understanding of the resiliency of thermal refugia with anticipated climate change (Fullerton et al. 2018; Frechette et al. 2018; Ficklin et al. 2018; Stevens and DuPont 2011; McCullough et al. 2009)." While the second part of this statement has apparently being done in the cited literature, the first part ("to validate temperature ranges by river feature at the watershed-scale") is done in this research. By connecting the two, you may get very valuable information about which restoration efforts are required to maintain a save passage for LCT. And such a quantitative analysis is also required to back-up all the suggested efforts listed just below this statement (P16, L14-24).

The literature cited in the introduction is generally species or system specific.  We expanded this section of the introduction (4th paragraph of the intro) and clarified which species existing thermal refugia literature refer to.  We have also rewritten the discussion to tie our results and findings into the literature.  It is not always meaningful to apply existing thermal refugia connectivity (rather than needed thermal refugia connectivity) or assume that thermal refugia needs of Lahontan cutthroat trout are the same as other species studied.  We make this clear in our revised discussion by highlighting how our research improves understanding of thermal refugia and how our method is a novel approach to analyze thermal refugia.

A second, slightly minor issue is that the author state that it is not possible to come up with maximum temperatures for the 50 and 300 m reaches of the TIR data due to the fact that part of the TIR data resembles the riparian zone. However, they also have the TIR summary points, which do report a maximum value for the 300 m reaches. So how are these maximum values obtained? At the same time I also wonder what causes the very small range in the TIR data compared to the DTS data (which is clearly visible in Fig. 7).

The summary points are explained page 7 ln 15-17: "Watershed Sciences, Inc. also provided summary point data, which are minimum, median, and maximum temperatures of 10 pixels from the middle of the stream.  Flight speed, image overlap, and river features determined which images to sample (Watershed Sciences Inc., 2012)."  This summary point method also explains why TIR data showed a smaller range of temperatures than the DTS data in Figure 7.

Line by line comments:
P1, L18: The abbreviation of DTS has not been defined yet
We added the acronym on P1, line 13.

P2, L22: rephrase: you cannot have fine spatial scales at point locations
We removed 'spatial' from this sentence.

P6, L6: How much time did it take to measure the whole stream? And how much would the temperature change over such a time period (you may get such an estimate from the temperature model).
We added another sentence (pg 7, ln 9-10): "Stream temperatures measured with temperature loggers warmed by 1 - 2 oC (average 1.6 oC) between 14:00 to 16:00 when TIR data were collected."

P6, L18: Was the average flow, the average over the TIR collection time?
We revised this sentence to clarify these were the average flows during the TIR data collection period.

P6 L32: Define which boundary conditions are needed
We specified that these are boundary condition streamflows.

P7, L19: Make clear that r refer to a 300m model reach and not to one of the two locations where DTS has been employed.
This refers to the DTS deployment site. To clarify, we changed the subscript for deployment site to s throughout the manuscript.

P8, L1: "One m extends…" ???
This was a typo. We changed it to read "For the 1 m comparison, we …".

P8, L12: I guess you mean the 'mean' instead of 'median'?
This is correct as we have written it. We used TIR summary points, which have data for minimum, median, and maximum stream temperatures. We then averaged the median values for each 300 m reach.

P8, L22-23: Do you mean outside the measured temperature range?
We corrected this sentence to say the 'measured temperature ranges'.

P10,L3: Explain what you mean with 'consistent temperatures'
We reworded this sentence to "Temperatures in the East Walker River changed more over time than over space."

P10,L5-6: This is Ti,r, isn't it? I suggest mentioning these parameters every time you report them, so the reader can easily go back to the methods to see which formula is used. Please do this throughout the manuscript. This will also help to see if all statistics parameters mentioned in section 3 are indeed used. Although I did not double check it, I don't recall to have seen values of Td,r.
We removed equations that were not used and included notation throughout the results section as recommended so that readers can easily go back to methods to see which formula is used.

P10,L25: cooling effect on what? It is indeed cooler in the drain than outside, but due to the limited length of observations downstream of the drain, it is hard to see any cooling effect here.
We changed wording to 'the cooler temperatures in the Wabuska Drain…'.
P11,L9: Do you mean that the temporal (e.g. daily) range of these features were large, or that they are locations with a distinctive lower/higher temperature than the mean spatial temperature of that specific range?

Good question – we meant the latter.  We changed this sentence to read (starting pg 10, ln 32) "In the East Walker River site, deep pools and reaches with large wood structures were river features with distinctively lower temperatures than the rest of the river.  In the mainstem Walker River, deep pools with riparian vegetation, beaver dams, and islands in the channel were river features that were cooler or warmer than spatially-averaged river temperatures."

P11,L13: "for one hour": I guess you mean "for a single point in time"?
We changed wording to 'a single point in time'.

P11,L24-25: Such a firm statement requires some proof, which is missing here. A few lines before it was stated that it MAY be due to such shallow groundwater contributions.
In fact, don't these shallow groundwater contributions, which are caused by irrigation, consist of the same water as the return flows (and thus with a similar temperature)?
We qualified the statement by saying " Thus, *monitoring suggests that* large diversions and return flows can create warm water conditions when active…"

Importantly, shallow groundwater and return flow contributions are from irrigation water; however return flow contributions are exposed to atmospheric conditions for longer (or a larger percentage of time once drained from fields) so temperatures may not be similar.

P12,L10: Maybe I misunderstood what has been compared here, but this statement implies that the minimum temperatures for all six 50 m reaches within a 300m reach should be the same. When looking at Fig. 5, this seems not to be the case with differences in minimum temperatures between the six 50m reaches of 1 or maybe 2 degrees C
The absolute minimum temperatures for the mainstem, East Walker, and West Walker Rivers do not change if lateral comparisons are for 50 m reaches or 300 m reaches.  We have revised wording of this section to clarify this point.  However, you bring up a good point that the average of the minimum temperatures vary for 50 m versus 300 m reaches.  We added a sentence to highlight how this differs based on scale of analysis (pg 12, ln 6-9): "However, minimum temperatures varied among 50 m river segments than made up each 300 m river segment (Fig. 5).  Thus, average minimum temperatures were 0.8 oC warmer when analyzing data at the 50 m scale than the 300 m scale.  This highlights the extent to which spatial temperature variability varies by the scale of analysis."

P13,L24-26: Please quantify this effect! In other words: what is the accuracy of this method?
The accuracy of our TIR data compared to temperature loggers was already included.  We moved it to the first paragraph of TIR stream temperature results to highlight it (P11, ln 4-7): "*TIR data were within 0.5 °C of iButton sensors, except for one location in the East Walker River where redundant sensors were 1.7 °C and 3.3 °C cooler than radiant TIR temperature, and one location in the West Walker River where an iButton was 1.1 °C cooler than radiant TIR temperature.  TIR measures water surface temperatures, so these discrepancies may have occurred where the river was not well mixed*."  It is outside the scope of this paper to quantify the effect of surface roughness, surface emissivity, surface reflection, variable background temperatures, turbidity, changes in viewing aspect, aircraft type, flight speed, wind gusts, and data collection time on TIR image and quality, but we would be remiss to not succinctly describe sources of data error in the limitations section.

P13,L31-32: I understand that this is outside the scope of this paper, but with some simple back-of-the-envelope calculations (e.g. a simple diffusion equation) it is possible to give an estimate or an upper limit of this stratification. This may also help to get an idea about the accuracy of the TIR data.

We disagree with this comment. Stratification is complex as it is a function of inflow velocities, orientation, slope, channel/pool geometry, as well as atmospheric influences including wind speed, air temperature, radiation penetration to the bed, bed conduction, groundwater inflows… To double check, we estimated stratification using pool geometry, thermocline heat transfer, and vertical diffusion. However, we had to make so many assumptions that stratification patterns and temperatures were not reliable estimates. Although we can come up with values, we have no reason to believe them and including them detracts rather than improves the paper.

P14,L7: "Future studies could collect data specifically to overlap in time and space": Please make clear what the gain is of doing so!
We changed this sentence to read (pg 14, ln 16-18): "Future studies could collect data specifically to overlap in time and space so that temperature distributions along the river are not affected by different years and sample periods."

P14,L18-19: "indicating that these methods complement each other": But it could also be that different periods result in different temperature distributions along the complete stream...
We added this thought to the manuscript. This sentence now reads (pg 14, ln 28-29): "… indicating that these methods complement each other, but also suggesting that different years may result in alternate temperature distributions along the river (Tables 2 and 3)."

P14,L32: "has poor aquatic habitat as a function of streamflow and stream temperature": What do you mean with this statement?
We revised this sentence to read (pg 15, ln 12-13): "Previous research has shown that the mainstem Walker River has low streamflows and warm stream temperatures that do not support LCT or other cold-water species …".

P15,L1-2: I am not familiar with those studies, but does this conclusion arises from results presented in this manuscript?
Or stated differently: Your results show that although the modelled stream water temperature may be too high, there are still places within each model reach that are colder (or cold enough). Can you subsequently use the findings of the studies listed here or in Line 4-7 of the introduction to indicate if these location for refugia are sufficient for LCT to survive?
We added the connectivity of thermal refugia in this basin (pg 13, ln 19-21): "The shortest distance between refugia, or cooler pockets of water, was 0.3 km, which was the spatial resolution of model reaches. The maximum distance between refugia was 37 km and occurred near Weber Reservoir in the mainstem Walker River. The mean distance between refugia was 2.8 km and the median distance was 0.9 km." We also rewrote the discussion to focus on how our research builds on previous work and highlight how our method quantifies thermal refugia connectivity using modeled and high-resolution measured data.

We also added a new 3rd paragraph to the discussion synthesizing temperature and thermal refugia needs for LCT.

P15,L11-12: Also here: Is it possible to connect your quantitative results with the studies described in L4-7 of the introduction. The same for L23-24 of this page
We have rewritten the discussion section and have done this. In particular, see the 3rd paragraph of the discussion.

P15,L19-20: I still don't understand what you mean: Is it a spatial temperature range that covers a 300 m modelling grid cell or is it a temporal range comparing day and night temperatures of the specific beaver dam?

We mean temperature variability over sampling event which were collected every 15 minutes. We reworded this section to read (pg 15, ln 5-7): "Beaver dams had especially high temperature variability, consistent with findings from Majerova et al. (2015) and Weber et al. (2017). A 7 oC temperature range was observed within a beaver dam in the mainstem Walker River during a DTS sampling event."

P15,L30-32: Are these values compared to the mean temperature of the 300m reach, or do they reflect the maximum range? In case of the latter you cannot simply say that the coldest temperature within a model reach is this much colder, while in case of the former you have to make explicit that in Fig. 9 you assume that the modelled temperature is the 'correct' average of the whole stream segment.

These values are added to the simulated temperature of the 300 m modeled reach. We clarified this on pg 13, ln 11-12: "Measured DTS and TIR temperature ranges from return flows, diversions, beaver dams, and seeps were added or subtracted to perfectly-mixed, 300 m modeled reach stream temperatures to estimate thermal refugia connectivity."

P16,L10-13: I don't understand why future research is needed for this: In the introduction you stated that this literature studied this effect. So why can you not use their results to say something about the survival changes of LCT for the Walker stream. Eventually you may come up with advice on where extra refugia are needed.
And to be more strict: such a quantitative analysis should be done first before you can suggest the list of restoration efforts listed in the next paragraph (P16,L14-24)

We revised this section to be more specific about future research needs. It now reads (pg 15, ln 32-34): "Additional work is needed to understand the resiliency of streamflows and thermal refugia with interannual variability and with anticipated climate change."

Fig. 2: In section 3.1, it is stated that ~400 m of cable is situated on either side of the river. So that means that the upper half of the plot should be more or less a mirror image of the lower half. So I think it is helpful if the flow direction is indicated in the graph, where the water is flowing to (or from) ~550m.

We added the flow direction to Figure 2 and labelled it as river right or river left.

Fig. 3: The purple dots indicate the borders of the 300m model reaches. However, the reach covered by the DTS cable is 400 (or 450). I understand there can be some kind of sinuosity in the cable, but a difference of 100 or 150 m seems rather large to me. To me it seems that the modelled stream reaches are too short and I am wondering which effect this has on the simulated stream water temperature.

Modeled stream reaches were delineated using 2011 river centerline. ArcGIS' split command was used to split the line into segments of equal length (Elmore et al. 2016; Elmore 2015). The RMS model represents 300 km of river with 999 nodes, thus each modeled reach is 300.3 meters. It is possible that the channel shifted between the 2011 channel layer used in the model and the 2015 channel observed during the DTS deployment. However, the suggestion that modeled reaches were too short or that modeling was sloppy is baseless.

Fig. 6: I don't understand the phrase "with the upstream-most river km on the left side of the x-axis". The same phrase is present in the caption of Fig. 7, and there I have the feeling that the authors mean that in the graphs the water is flowing from left to right.

[revised manuscript text omitted]
 and the modeled reach that contained those features was identified.  We added or subtracted measured temperature ranges to  model at georeferenced river features to estimate spatial variability missing in model output.  Diversion and return flow locations were identified in 2012 by the Walker Basin Project (Tim Minor, pers.comm, 2012).  Seeps were identified during TIR surveys from cooler stream temperatures that could not be attributed to shadows, cutbanks, or vegetation (Watershed Sciences Inc. 2011).  We used seep locations identified during the winter TIR flight completed on November 16-17, 2011 because temperature differences were more obvious than the summer flight and some of the locations with groundwater seeps in the winter were dry during the summer flight (Watershed Sciences Inc., 2011; 2012).  We applied the temperature range observed at seeps during the summer 2012 TIR flight (Watershed Sciences Inc. 2012).

Beaver are native to the Walker Basin (Gibson and Olden, 2014) and beaver dams were identified using 2012 and 2013 Google Earth aerial imagery (Google Earth Pro, 2018).  We included beaver dams that spanning the channel.  Often turbulence was observed below the dam and sometimes crowdsourced photos added images of the beaver dams from the ground.  We relied primarily on 2012 imagery, unless it was unavailable or of poor quality, when 2013 aerial imagery was used.  2012 and 2013 were dry years, and beaver dams are more abundant in the Walker River during dry years, when high flow events that limit beavers ability to dam across the stream channel are reduced (Nevada Department of Wildlife, 2016).

**4 Results**

**4.1 DTS Measured Stream Temperatures and Ranges**

Average RMSE between calibrated DTS data and the three reference temperatures was 0.09 °C and 0.15 °C for the East Walker River and mainstem Walker River DTS sites, respectively (Table S1).  Average DTS error for both sites was also within the 0.5 °C precision of the iButtons.  There were no significant residual trends in errors for the mainstem Walker River (Table S2 and Fig. S1).

The East Walker River changed more through time than through space (Fig. 2).  The deployment period minimum stream temperature $(Tmin_i)$ was 16.7 °C and maximum temperature $(Tmax_i)$ was 24.9 °C (Table 2).  Maximum temperatures were measured in a straight, homogenous, unshaded section (Fig. 3).  Stream temperature range for 15 minute collection events $(R_i)$ varied from a

minimum of 0.5 °C to a maximum of 2.0 °C for the deployment period, with an average (RBAR$_{p,s}$) of 1.0 °C. A shaded backwater eddy and pools with overhanging shrubs and tall cottonwoods were river features with increased thermal heterogeneity in the East Walker River (Fig. 3).

**Figure 2: Stream temperatures measured for the length of the DTS cable at East Walker River (a) and mainstem Walker River (b) DTS sites. Wabuska Drain, which was not flowing but had standing water during sampling, is located at cable distance 110-175 m in the mainstem Walker River site (b).**

**Table 2: Daily stream temperatures and ranges for DTS deployments in the East Walker River (11:15 on 6/19/15 to 9:45 on 6/23/15) and mainstem Walker River (14:15 on 6/25/15 to 12:30 on 6/30/19). Daily stream temperatures and ranges for DTS deployment reaches in the East Walker and mainstem Walker Rivers. Data collection began in the afternoon on deployment days, June 19th and 25th, and ended in the morning of June 23rd and 30th.**

**Figure 3: East Walker River daily maximum stream temperatures on June 21, 2015 at 5:30 pm with insets showing details of spatial temperature variability. Modeled reach points represent the division between 300 m modeled reaches.**

Stream temperatures varied spatially throughout the mainstem DTS site, visualized as longitudinal color striations at different locations in Figure 2b. Average reach deployment site temperature (TBAR$_{p,s}$) was 25.2 °C, not including the Wabuska Drain segment (Table 2, excluding distance 110 – 175 m in Fig. 2b). Maximum stream temperature (Tmax$_i$) was 32.9 °C. The average reach temperature range for the deployment (RBAR$_{p,s}$) was 2.7 °C, with a minimum deployment site reach temperature range (Rmin$_i$) of 1.1 °C and a maximum site reach temperature range (Rmax$_i$) of 7.0 °C. Average DTS stream temperatures (TBAR$_{
[revised manuscript text omitted]
. Summer home ranges of Colorado River cutthroat trout have a median of 0.2 km (Young, 2004) and Bonneville cutthroat trout do not move more than 0.5 km during summer. This suggests that the existing network of thermal refugia in the lower Walker river may be adequate for LCT to move between spawning and lake habitats (following lake restoration), but are unlikely to provide refugia necessary for summer habitat.

From a broader perspective, thisOur research contributes to literature describing thermal refugia networks and how they may be included for watershed management (Isaak et al., 2012; Sutton et al., 2007). River features like diversions, return flows, and beaver dams provide temperature variability, and often, thermal refugia for cold water species like LCT. Fine spatial and temporal resolution stream temperature monitoring paired with watershed-scale modelling indicates that the distance between refugia varied from 0.3 to 37 km, closer together than the 5.7 to 49.4 km demonstrated by Fullerton et al. (2018) in the Pacific Northwest. Stream temperatures suggest that if LCT and other native fish have not migrated through warm reaches by summer, they must shelter in refuges to thermoregulate body temperature (Frechette et al., 2018). Since stream temperatures neared or exceeded LCT temperature thresholds for extended periods, foraging habitat near to thermal refugia are likely needed to maintain body temperatures (Pepino et al., 2015). However, trout use of thermal refugia may vary, as availability of refugia change with streamflow and weather conditions, and as trout habitat needs vary with life stage (Frechette et al., 2018; Dugdale et al., 2013).

Future research is needed to reduce uncertainty and validate the large temperature ranges observed for return flows, diversions, beaver dams, and seeps. Additional work is also needed to quantify the distance between thermal refugia and foraging habitats in this system (Pepino et al., 2015), the maximum distance between refugia for LCT to move between spawning and summer habitats (Shrank and Rahel, 2004), and to improve understanding of the resiliency of streamflows and thermal refugia with anticipated climate change (McCullough et al. 2009; Ficklin et al., 2018; Null and Prudencio, 2016). DTS and TIR stream temperature measurements bound temperature variability and can be used with simulation models in other watersheds to identify river features that provide thermal refugia, create temperature barriers, and inform restoration. Our approach may also be used 
[revised manuscript text omitted]